# Interpreting Reinforcement Policies Through Local Behaviors

## Abstract

Many works in explainable AI have focused on explaining black-box classification models. Explaining deep reinforcement learning (RL) policies in a manner that could be understood by domain users has received much less attention. In this paper, we propose a novel perspective to understanding RL policies based on identifying important states from automatically learned meta-states. The key conceptual difference between our approach and many previous ones is that we form meta-states based on locality governed by the expert policy dynamics rather than based on similarity of actions, and that we do not assume any particular knowledge of the underlying topology of the state space. Theoretically, we show that our algorithm to find meta-states converges and the objective that selects important states from each meta-state is submodular leading to efficient high quality greedy selection. Experiments on three domains (four rooms, door-key and minipacman) and a carefully conducted user study illustrate that our perspective leads to better understanding of the policy. We conjecture that this is a result of our meta-states being more intuitive in that the corresponding important states are strong indicators of tractable intermediate goals that are easier for humans to interpret and follow.

## 1 Introduction

Deep reinforcement learning has seen stupendous success over the last decade with superhuman performance in games such as Go (Silver et al., 2016), Chess (Silver et al., 2018) as well as Atari benchmarks (Mnih et al., 2015). With increasing superior capabilities of automated (learning) systems, there is a strong push to understand the reasoning behind their decision making. One motivation is for (professional) humans to improve their performance in these games (Rensch, 2021). An even deeper reason is for humans to be able to trust these systems if they are deployed in real life scenarios (Gunning, 2017). Safety, for instance, is of paramount importance in applications such as self-driving cars or deployments on unmanned aerial vehicles (UAVs) (Garcia & Fernandez, 2015). The General Data Protection Regulation (Yannella & Kagan, 2018) passed in Europe demands that explanations need to be provided for any automated decisions that affect humans. While various methods with different flavors have been provided to explain classification models (Ribeiro et al., 2016; Lundberg & Lee, 2017; Lapuschkin et al., 2016; Dhurandhar et al., 2018) and evaluated in application-grounded manner (Doshi-Velez & Kim, 2017; Dhurandhar et al., 2017), the exploration of different perspectives to explain reinforcement learning (RL) policies has been limited and user study evaluations comparing methods are rarely employed in this space.

In this paper, we provide a novel perspective to produce human understandable explanations with a task-oriented user study that evaluates which explanations help users predict the behavior of a policy better. Our approach involves two steps: 1) learning meta-states, i.e., clusters of states, based on the dynamics of the policy being explained, and 2) within eat meta-state, identifying states that act intermediate goals, which we refer to as *strategic states*. Contrary to the global nature of recent explainability works in RL (Topin & Veloso, 2019; Sreedharan et al., 2020; Amir & Amir, 2018), our focus is on local explanations; given the current state, we explain the policy moving forward within a fixed distance from the current state. This key distinction allows us to consider richer state spaces (i.e., with more features) because the locality restricts the size of the state space we consider, as will be demonstrated. It is also important to recognize the difference from bottlenecks (Menache et al., 2002; Simsek & Barto, 2004) which are *policy-independent* and learned by approximating the state space with randomly sampled trajectories; rather than help explain a policy, bottlenecks are

used to *learn* efficient policies such as through hierarchical RL (Botvinick et al., 2008) or options frameworks (Ramesh et al., 2019). Strategic states, however, are learned with respect to a policy and identified without assuming access to the underlying topology.

An example of this is seen in Figure 1a. Each position is a state and a meta-state is a cluster of possible positions (states sharing a color/marker). Within each meta-state, we identify certain states as *strategic states* (shown with larger markers), which are the intermediate states that moving towards will allow the agent to move to another meta-state and get closer to the goal state, which is the final state that the agent wants to get to. In Figure 1a, each room is (roughly) identified as a meta-state by our method with the corresponding doors being the respective strategic states. Topology refers to the graph connecting states to one another; our method only has access to the knowledge of which states are connected (through the policy), whereas reinforcement learning algorithms might have access to properties of the topology, e.g., the ability to access similar states using successor representations (Machado et al., 2018). In Figure 1, the topology is a graph connecting the different positions in each room or the doors connecting one room to another.

A key conceptual difference between our approach and others is that other methods aggregate insight (i.e. reduce dimension) as a function of actions (Bastani et al., 2018) or formulas derived over factors of the state space (Sreedharan et al., 2020) to output a policy summary, whereas we aggregate based on locality of the states determined by the expert policy dynamics and further identify strategic states based on these dynamics. Still other summarization methods simply output simulated trajectories deemed important (Amir & Amir, 2018; Huber et al., 2021) as judged by whether or not the action taken at some state matters. We use the term *policy dynamics* to refer to state transitions and high probability paths. We use the term dynamics because this notion contrasts other methods that use actions to explain what to do in a state or to identify important states; strategic states are selected according to the trajectories that lead to them, and these trajectories are implicitly determined by the policy.

The example in Figure 1 also exposes the global view of our explanations when the state space is small because local approximations of the state space are not needed. We show that this perspective leads to more understandable explanations; aggregating based on actions, while precise, are too granular a view where the popular idiom *can't see the forest for the trees* comes to mind. We conjecture that the improved understanding is due to our grouping of states being more intuitive with strategic states indicating tractable intermediate goals that are easier to follow. An example of this is again seen in Figures 1b and 1c, where grouping based on actions for interpretability or for efficiency leads to less intuitive results (note that Figure 1c replicates Figure 4b from (Abel et al., 2019)). A more detailed discussion of this scenario can be found in section 5, where yet other domains have large state spaces and require strategic states to explain local scenarios.

As such, our main contributions are two-fold:
1. We offer a novel framework for understanding RL policies, which to the best of our knowledge, differs greatly from other methods in this space which create explanations based on similarity of actions rather than policy dynamics. We demonstrate on three domains of increasing difficulty.
2. We conduct a task-oriented user study to evaluate effectiveness of our method. Task-oriented evaluations are one of the most thorough ways of evaluating explanation methods (Doshi-Velez & Kim, 2017; Lipton, 2016; Dhurandhar et al., 2017) as they assess simulatability of a complex AI model by a human, yet to our knowledge, have rarely been used in the RL space.

## 2 NOTATION

We use the following notations. Let $\mathcal{S}$ define the full state space and $s \in \mathcal{S}$ be a state in the full state space. Denote the expert policy by $\pi_E(\cdot, \cdot) : (\mathcal{A}, \mathcal{S}) \to \mathbb{R}$ where $\mathcal{A}$ is the action space. The notation $\pi_E \in \mathbb{R}^{|\mathcal{A}| \times |\mathcal{S}|}$ is a matrix where each column is a distribution of actions to take given a state (i.e., the policy is stochastic). Note that we assume a transition function $f_E(\cdot, \cdot) : (\mathcal{S}, \mathcal{S}) \to \mathbb{R}$ that defines the likelihood of going from one state to another state in one jump by following the expert policy.

Let $\mathcal{S}_\phi = \{\Phi_1, ..., \Phi_k\}$ denote a meta-state space of cardinality $k$ and $\phi(\cdot) : \mathcal{S} \to \mathcal{S}_\phi$ denote a meta-state mapping such that $\phi(s) \in \mathcal{S}_\phi$ is the meta-state assigned to $s \in \mathcal{S}$. Denote $m$ strategic states of meta-state $\Phi$ by $G^\Phi = \{g_1^\Phi, ..., g_m^\Phi\}$ where $g_i^\Phi \in \mathcal{S} \ \forall i \in \{1, ..., m\}$.

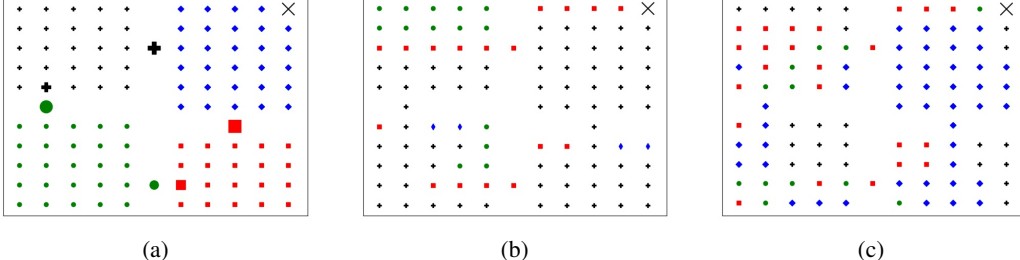

(a)    (b)    (c)

Figure 1: Illustrations of our SSX (a), VIPER-D (b), and abstract states used for compression (c) methods based on an expert policy for the Four Rooms game with neither having information about the underlying topology of the state space. Colors/Shapes denote different meta-states/clusters. The black **X** in the upper right is the goal state. SSX clusters the four rooms exactly with strategic states denoted by larger markers, where the biggest marker implies the priority strategic state. SSX explains that the expert policy will head towards the open doors in each room preferring the door that leads to the room with the goal state. VIPER-D clusters states by action (black/plus=up, green/circle=down, blue/diamond=left, red/square=right) based on the full (discrete) state space, rather than samples, since it is tractable here. The compressed state space in (c) is also a function of the experts (conditional) action distribution. Clusters in (b) and (c) are scattered making it challenging for a human to understand any policy over clusters.

## 3 METHOD

We now describe our algorithm, the Strategic State eXplanation (SSX) method, which involves computing shortest paths between states, identifying meta-states, and selecting their corresponding strategic states. However, we first define certain terms. Recall that all paths discussed below are based on transitions dictated by an expert policy because we want to explain the policy; the well-known concept called bottlenecks are identified from paths generated as random walks through the state space and are meant to help learn policies rather than explain them.

**Maximum likelihood (expert) paths:** One criterion used below is that two states in the same meta-state should not be far away from each other. The distance we consider is the most likely path from state $s$ to state $s'$ under $\pi_E$. Consider a fully connected, directed (in both directions) graph where the states are vertices and an edge from $s$ to $s'$ has weight $-\log f_E(s, s')$. By this definition, the shortest path is also the maximum likelihood path from $s$ to $s'$. Denote by $\gamma(s, s')$ the value of this maximum likelihood path and $\Gamma \in \mathbb{R}^{|\mathcal{S}| \times |\mathcal{S}|}$ a matrix containing the values of these paths for all pairs of states in the state space. $\Gamma$, along with a predecessor matrix $P$ that can be used to derive the shortest paths, can be computed using Dijkstra's shortest path algorithm in $O(|\mathcal{S}|^2 \log |\mathcal{S}|)$ because all edge weights are non-negative. Section 3.4 below discusses how our algorithm is applied with a large state space. Note that computation of $\Gamma$ means that SSX requires access to a policy simulator for $\pi_E$, and in practice, might require simulation for estimation when $\Gamma$ cannot be computed exactly. This is a common requirement among related explanation methods, e.g., in order to simulate important trajectories Amir & Amir (2018) or samples to train a decision tree Bastani et al. (2018), that are discussed below in Section 4.

**Counts of Out-paths:** Another criterion used below for assigning states to meta-states is that if state $s$ lies on many of the paths between one meta-state $\Phi_i$ and all other meta-states, then $s$ should be assigned the meta-state $\Phi_i$, i.e., $\phi(s) = \Phi_i$. We define, for fixed state $s$ and its assigned meta-state $\phi(s)$, the number of shortest paths leaving $\phi(s)$ that $s$ lies on. Denote $T(s, s')$ as the set of states that lie on the maximum likelihood path between $s$ and $s'$, i.e., the set of states that define $\gamma(s, s')$. Then $1[s \in T(s', s'')]$ is the indicator of whether state $s$ lies on the maximum likelihood path between $s'$ and $s''$, and we compute the count of the number of such paths for state $s$ and meta-state $\phi(s)$ via

$$C(s, \phi(s)) = \sum_{\substack{s' \neq s, \\ \phi(s') = \phi(s)}} \sum_{\substack{s'': \\ \phi(s'') \neq \phi(s)}} 1[s \in T(s', s'')]. \tag{1}$$

$C(s, \phi(s))$ can be computed for all $s \in \mathcal{S}$ in $O(|\mathcal{S}|^2)$ by iteratively checking if predecessors of shortest paths from each node to every other node lie in the same meta-state as the first node on the

path. Note this predecessor matrix was already computed for matrix $\Gamma$ above. One may also consider the likelihood (rather than count) of out-paths by replacing the indicator in eq. (1) with $\gamma(s', s'')$.

## 3.1 LEARNING META-STATES

We seek to learn meta-states that balance the criteria of having high likelihood paths within the meta-state and having many out-paths from states within the meta-state. This is accomplished by minimizing the following objective for a suitable representation of $s$, which in our case is the eigen-decomposition of the Laplacian of $\Gamma$:

$$\underset{\mathcal{S}_\phi}{\mathrm{argmin}} \sum_{\Phi \in \mathcal{S}_\phi} \sum_{s \in \Phi} \left[ (s - c_\Phi)^2 - \eta C(s, \Phi) \right] \tag{2}$$

where $c_\Phi$ denotes the centroid of the meta-state $\Phi$ and $\eta > 0$ balances the trade-off between the criteria. Note that we are optimizing $\mathcal{S}_\phi$ over all possible sets of meta-states. Other representations for $s$ and functions for the first term could be used, but our choice is motivated from the fact that such formulations are nostalgic of spectral clustering (Shi & Malik, 2000) which is known to partition by identifying well-connected components, something we strongly desire. Our method for solving eq. (2) is given by algorithm 1 and can be viewed as a regularized version of spectral clustering. In addition to clustering a state with others that it is connected to, the regularization term pushes a state to a cluster, even if there are only a few connections to the cluster, if the policy dictates that many paths starting in the cluster go through that state.

---

**Algorithm 1:** Meta-state function $\mathsf{MS}(\mathcal{S}, \mathcal{A}, \pi_E, \Gamma, k, \epsilon_\phi, \eta)$

1) Get eigen representation of each state $s$ from eigen decomposition of the Laplacian of $\Gamma$
2) Randomly assign states $s \in \mathcal{S}$ to a meta-state in $\mathcal{S}_\phi = \{\Phi_1, ..., \Phi_k\}$ and compute centroids
   $c_1, ..., c_k$ for meta-states
3) $\xi^{\mathrm{cur}} =$ current value of objective in eq. (2)
**do**
    4) $\xi^{\mathrm{prev}} = \xi^{\mathrm{cur}}$
    5) Reassign states $s$ to the meta-states based on smallest value of $(s - c_\Phi)^2 - \eta C(s, \Phi)$
    6) Compute centroids $c_1, ..., c_k$ for meta-states based on current assignment
    7) $\xi^{\mathrm{cur}} =$ current value of objective in eq. (2)
**while** $|\xi^{cur} - \xi^{prev}| \geq \epsilon_\phi$;
**Output:** Meta-states $\{\Phi_1, ..., \Phi_k\}$

---

**Algorithm 2:** Strategic State function $\mathsf{SS}(\mathcal{S}_\phi, \Gamma, \epsilon_g)$. Finds Strategic States with Greedy Selection (w.l.o.g. assume meta-state $\Phi_k$ contains the goal state).

**for** $i = 1$ *to* $k - 1$ **do**
    1) Let $\xi^{\mathrm{cur}} = 0$ and $G_{\Phi_i} = \emptyset$
    **do**
        2) $\xi^{\mathrm{prev}} = \xi^{\mathrm{cur}}$
        3) $G_{\Phi_i} = G_{\Phi_i} \cup g$ where $g = \underset{s \in \Phi_i \backslash G_{\Phi_i}}{\mathrm{argmax}}$ of eq. (3) given the current strategic states $G_{\Phi_i}$
        4) $\xi^{\mathrm{cur}} =$ evaluate eq. (3) with $G_{\Phi_i}$
    **while** $|\xi^{cur} - \xi^{prev}| \geq \epsilon_g$;
**end**
5) $G_{\Phi_k} = g$, where g denotes the goal state of the expert policy
**Output:** Strategic states corresponding to each meta-state $\{G_{\Phi_1}, ..., G_{\Phi_k}\}$

---

## 3.2 IDENTIFYING STRATEGIC STATES

Next, strategic states must be selected for each meta-state. Assume that $g_1^\Phi, ..., g_m^\Phi \in \mathcal{S}$ are $m$ strategic states for a meta-state $\Phi$ that does not contain the target state. Our method finds strategic states by solving the following optimization problem for some $\lambda > 0$:

$$G_\Phi^{(m)} = \underset{g_1^\Phi, ..., g_m^\Phi}{\mathrm{argmax}} \sum_{i=1}^{m} C(g_i^\Phi, \Phi) - \lambda \sum_{i=1}^{m-1} \sum_{j=i+1}^{m} \max \left( \gamma(g_i^\Phi, g_j^\Phi), \gamma(g_j^\Phi, g_i^\Phi) \right) \tag{3}$$

The first term favors states that lie on many out-paths from the meta-state, while the second term favors states that are far from each other. Thus, the overall objective tries to pick states that go to different highly rewarding parts of the state space from a particular meta-state, while also balancing the selection of states to be diverse (i.e., far from each other). The objective in eq. (3) is submodular as stated next (proof in appendix) and hence we employ greedy selection in algorithm 2. Note that for the meta-state that contains the target state, the target state itself is its only strategic state.

**Proposition 1.** *The objective to find strategic states in equation (3) is submodular.*

### 3.3 STRATEGIC STATE EXPLANATION (SSX) METHOD

Our full method is detailed as follows. First, the maximum likelihood path matrix $\Gamma$ is computed. Then, algorithm 1 tries to find meta-states that are coherent w.r.t. the expert policy, in the sense that we group states into a meta-state if there is a high likelihood path between them. Additionally, if many paths from states in a meta-state go through another state, then the state is biased to belong to this meta-state. Finally, algorithm 2 selects strategic states by optimizing a trade-off between being on many out-paths with having a diverse set of strategic states. Code is provided in the Supplement.

### 3.4 SCALABILITY

Given our general method, we now discuss certain details that were important for making our algorithm practical when applied to different domains. SSX is applied in Section 5 to games with state spaces ranging from small to exponential in size. SSX is straightforward for small state spaces as one can pass the full state space as input, however, neither finding meta-states nor strategic states would be tractable with an exponential state space. One approach could be to compress the state space using VAEs as in (Abel et al., 2019), but as shown in Figure 1c, interpretability of the state space can be lost as there is little control as to how states are grouped. Our approach is to use local approximations to the state space; given a starting position, SSX approximates the state space by the set of states within some $N > 0$ number of moves from the starting position. In this approach, Algorithms 1 and 2 are a function of $N$, i.e., increasing $N$ increases the size of the approximate state space which is passed to both algorithms. One can contrast our approach of locally approximating the state space with that of VIPER (Bastani et al., 2018) which uses full sample paths to train decision trees. While the number of states in such an approximation is $M^N$, where $M$ is the number of possible agent actions, the actual number of states in a game such a pacman is much smaller in practice. Indeed, while pacman has 5 possible actions, the growth of the state space in our approximation as $N$ increases acts similar to a game with between 2-3 actions per move because most states in the local approximation are duplicates due to both minipacman and the ghost going back and forth. See Figure 5 in the Appendix, where other practical considerations are also discussed.

## 4 RELATED WORK

While a plethora of methods are proposed in XAI (Ribeiro et al., 2016; Lundberg & Lee, 2017; Lapuschkin et al., 2016; Dhurandhar et al., 2018), we focus on works related to RL explainability and state abstraction, as they are most relevant to our current endeavor.

Most global RL methods summarize a policy using some variation of state abstraction where the explanation uses aggregated state variables that group actions (Bastani et al., 2018) using decision trees or state features (Topin & Veloso, 2019) using importance measures, or such that an ordering of formulas based on features is adhered to (Sreedharan et al., 2020). These approaches all intend to provide a global summary of the policy. Other summaries output trajectories deemed important according to importance measures (Amir & Amir, 2018; Huber et al., 2021) or through imitation learning (Lage et al., 2019), or train finite state representations to summarize a policy with an explainable model (Danesh et al., 2019; 2021). Visualization techniques combined with saliency have been used to either aggregate states and view the policy from a different perspective (Zahavy et al., 2016) or create a trajectory of saliency maps (Greydanus et al., 2018). Further, other works try to find state abstractions or simplify the policy (Abel et al., 2019; Paul et al., 2019; Liang et al., 2016), and one should not confuse these works with those seeking explainability. State abstraction in these works is used to compress the state space so that simpler policies can be used; the compressed state space is not intepretable as seen in Figure 1c.

Turning towards local explanation methods, some works focus on self-explaining models (Mott et al., 2019) where the policy has soft attention and so can indicate which (local) factors it is basing its decision on at different points in the state space. Yau et al. (2020) learns a *belief map* concurrently during training which is used to explain locally by predicting the future trajectory. Interestingly, there are works which suggest that attention mechanisms should not be considered as explanations (Jain & Wallace, 2019). These directions focus on learning an inherently explainable model rather than explaining a given model which is our goal. Other works use local explanation methods to explain reasons for a certain action in a particular state (Olson et al., 2021; Madumal et al., 2020). These are primarily contrastive where side information such as access to the causal graph may be assumed. Our approach besides being methodologically different, also differs conceptually from these, where we form meta-states based on policy dynamics and then identify (strategic) states through which many policy-driven paths cross.

There are also program synthesis-type methods (Verma et al., 2018; Inala et al., 2020) that learn syntactical programs representing policies, which although more structured in their form, are typically not amenable to lay users. There are also methods in safe RL that try to uncover failure points of a policy (Rupprecht et al., 2020) by generating critical states. Another use of critical states, defined differently by how actions affect the value of a state, is to establish trust in a system (Huang et al., 2018). There is also explainability work in the markov decision processes literature which focus on filling templates according to different criteria such as frequency of state occurrences or domain knowledge (Khan et al., 2009; Elizalde et al., 2009). A more elaborate discussion of these and other methods can be found in (Alharin et al., 2020), all of which unequivocally are different from ours.

## 5 EXPERIMENTS

This section illustrates the Strategic State eXplanation (SSX) method on three domains: four rooms, door-key, and minipacman. These domains represent different reinforcement learning (RL) regimes, namely, 1) non-adversarial RL with a small state space and tabular representation for the policy, 2) non-adversarial RL, and 3) adversarial RL, the latter two both with a large state space and a deep neural network for the policy. These examples illustrate how strategic states can aid in understanding RL policies. Experiments were performed with 1 GPU and up to 16 GB RAM. The number of strategic states was chosen such that additional strategic states increased the objective value by at least 10%. The number of meta-states was selected as would be done in practice, through cross-validation to satisfy human understanding. Details about environments are in the Appendix, along with additional experiments illustrating faithfulness and consistency of SSX.

**Four Rooms:** The objective of Four Rooms is move through a grid and get to the goal state (upper right corner). The lack of a marker in a position represents a wall. Our grid size is $11 \times 11$. The state space consists of the current position of a player and the policy is learned as a tabular representation, since the state space is not too large, using Value Iteration (Martino & Mostofsky, 2016).

SSX is displayed in Figure 1a with settings that learn four meta-states. Clustering the states using algorithm 1 according to the policy dynamics (i.e. maximum likelihood path matrix $\Gamma$) results in an (almost) perfect clustering of states according to the rooms. X's denote strategic states learned in each meta-state, with a larger X corresponding to the first strategic state found. Clearly either door in blue, green or red rooms could lead to the goal state in the upper right corner (large yellow diamond), but it is important to note that higher valued strategic states in the red and blue rooms are those that lead directly to the yellow room where the goal state is located.

Figure 1b illustrates the results of VIPER-D which is our implementation of VIPER (Bastani et al., 2018). The explanation is illustrated using different colors per action which effectively offers the rules of the decision tree. While an explanation based on rules can be informative in continuous state spaces (as demonstrated in (Bastani et al., 2018)), such rules applied to a discrete state space as done here may lead to confusion, e.g., groups of green states are split by yellow states in the left two rooms and allow for an optimal policy but it is not clear how to describe the cluster of states in which to take each action. Figure 1c illustrates the difference between explainability and compression. The purpose of (Abel et al., 2019) is to learn abstract states upon which a proxy policy can be learned more efficiently that replicates the original expert policy on the full state space. The lack of interpretability of the abstract states is not of concern in that context.

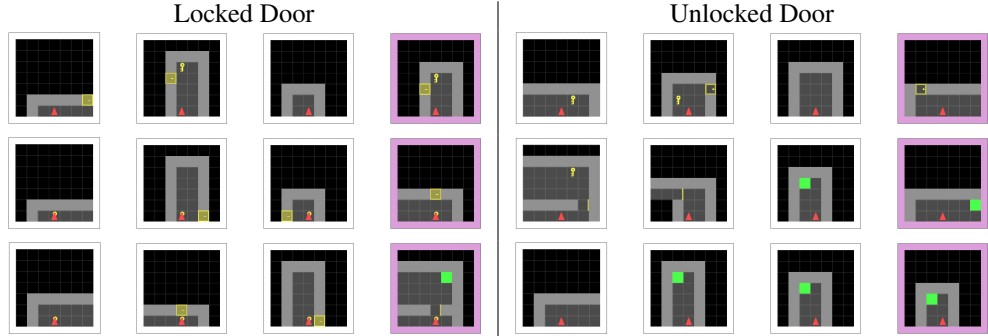

Figure 2: Illustration of our SSX method on Door-Key. Policies were trained on two different environments: Locked Door and Unlocked Door. Each row corresponds to a meta-state and strategic state (outlined in pink) from running SSX starting at a different number of moves into the same path (one path for completing the task in each of the two environments).

**Door-Key:** Door-Key is another non-adversarial game, but differs from Four Rooms because the state space is exponential in the size of the board. The policy is learned as a convolutional neural network (CNN) with three convolutional and two linear layers. In this game, one must navigate from one room through a door to the next room and find the goal location to get a reward. Policies are trained under two scenarios. In the first scenario, there is a key in the first room that must be picked up and used to unlock the door before passing through. In the second scenario, the door is closed but unlocked, so one does not need to first pick up the key to open the door.

SSX is run with local approximations to the state space with the maximum number of steps set to 6 as discussed in Section 3.4. Results are shown in Figure 2. The state space is a $7 \times 7$ grid reflecting the forward facing perspective of the agent. Walls are light gray and empty space visible to the agent is dark gray. Grid positions blocked from view by walls are black. The scenes in Figure 2 are exactly what a user sees. To better understand why the scenes do not appear easily connected, consider the first two states in the first row - the only difference from the first state is that the agent has changed directions. When facing the wall, the agent's view only includes the three positions to the right and one position to the left. All positions on the other side of the wall are not visible to the agent, which is depicted as black. When the agent changed directions (row 1, column 2), many more positions in the room become visible to the agent.

In Figure 2, a sample path was generated using each policy. SSX was run at three different states along these paths, and one meta-state and corresponding strategic state (outlined in pink) from each SSX explanation is displayed. The three strategic states for the locked door environment correspond to the agent looking for the key (row 1), getting the key (row 2), and using it to open the door (row 3). The three strategic states for the unlocked door environment correspond to the agent immediately looking for the door (row 1), made it through the door (row 2), and moving toward the target (row 3).

For intuition on how a human would use these explanations, consider the cluster in row 1 for the Locked Door. Comparing the first three states in the cluster to the strategic state, a human sees that in this cluster, the policy is suggesting to face the key and move closer to it. As this is a local explanation, it is limited by the initial state being explained as to how close one get to the key. The cluster in row 1 for the Unlocked Door shows that the policy at these states is to face the door. Perhaps facing the door within a certain distance is how the policy breaks down the ultimate strategy. While one might wonder why the strategy is not to get closer to the door (e.g., move up from the second column), recall that the strategic state is explaining the policy and not human intuition.

Lastly, note that for the Unlocked Door, the third state is the same in rows 2 and 3. The rows correspond to explanations for two different initial states, but it is very possible that the same state is encountered in trajectories from each initial state and thus appears in multiple explanations as seen here. Such occurrences further illustrate that SSX explanations are local to an initial state.

**Minipacman:** Minipacman is a small version of the classic Pacman game. This game differs from Door-Key with the addition of an adversary - the ghost. The state space is again exponential in the size of the board and the policy is learned as a convolutional neural network with two convolutional and two linear layers. Two policies are trained with different objectives. The first objective, denoted

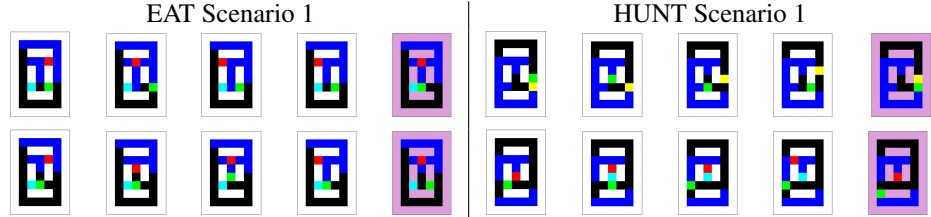

Figure 3: Illustration of our SSX method on minipacman. Two policies, EAT and HUNT, are displayed. Two clusters, one per row, are shown as part of the SSX result. The last board with pink background is a strategic state for each cluster. The color scheme is as follows: green = pacman, red = ghost, yellow = edible ghost, cyan = pill, blue = food, black = food eaten, white/pink=wall.

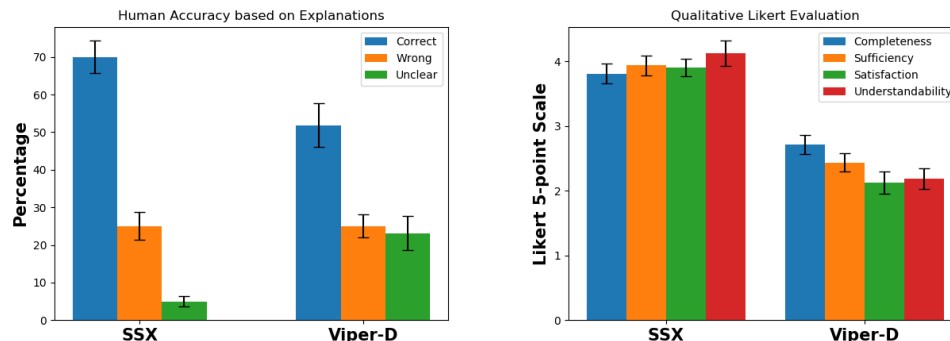

Figure 4: Above (left) we see the percentage (human) accuracy in predicting if the expert policy is Eat or Hunt based on SSX and Viper-D. As can be seen users perform much better with SSX with difference in performance being statistically significant (paired t-test p-value=0.01). Above (right) we see a 5-point Likert scale (higher better) for four qualitative metrics used in previous studies (Madumal et al., 2020). Here too the difference is statistically significant for all four metrics (p-values are all less than $2 \times 10^{-5}$). Error bars are 1 std error.

EAT, is for minipacman to eat all the food with no reward for eating the ghost. The second objective, denoted HUNT, is for minipacman to hunt the ghost with no reward for eating food.

SSX is again run with local approximations to the state space with the maximum number of steps set to 8. The state space is a $10 \times 7$ grid reflecting where food, pacman, a ghost, and the pill are located. Figure 3 displays one sample scenario under both the EAT and HUNT policies, with two meta-states and corresponding strategic states highlighted in pink. The two strategic states of EAT Scenario 1 show pacman eating the food (row 1) but then avoiding the ghost and ignoring the pill (row 2). In HUNT Scenario 1, pacman is either directly moving towards the ghost after having eaten the pill (row 1) or heading away from the pill while the ghost is near it (row 2). Two additional scenarios for EAT and HUNT can be found in the Appendix. An additional experiment with a baseline motivated by Amir & Amir (2018) appears in the Appendix.

# 6 USER STUDY

We designed a user study to evaluate the utility of our approach relative to the more standard approach of explaining based on grouping actions. While SSX has thus far been used to give users local explanations about particular scenarios, we use it here to gain insight as to the general goal of a policy because the relevant explanations to compare with are global; as previously discussed, other local literature is about learning inherently explainable models rather than explaining a fixed model or learning contrastive explanations which should be used complementary to our methods. The global applicability of SSX can also be seen as another advantage. As with Four Rooms, we again compare with our implementation of VIPER – a state-of-the-art explanation method for reinforcement learning policies – with a visual output tailored for the discrete state space. We do not compare with methods that output trajectories (Amir & Amir, 2018) as they require estimating Q-values to determine state

importance which relies on sensitive hyperparameter tuning. Among explanation methods, Viper makes for the best comparison as it requires a similar amount of human analysis of the explanation (by observing states), and while meant for global explainability, also gives local intuitions, as opposed to other global methods. The utility of each approach is measured through a task posed to study participants: users must guess the intent of the expert policy based on provided explanations which are either output by SSX or Viper-D. Such a task oriented setup for evaluation is heavily encouraged in seminal works on XAI (Doshi-Velez & Kim, 2017; Lipton, 2016; Dhurandhar et al., 2017).

**Setup:** We use the minipacman framework with the EAT and HUNT policies trained for Section 5 and each question shows either an SSX explanation or Viper-D explanation and asks the user "Which method is the explanation of type A (or B) explaining?" to which they must select from the choices Hunt, Eat, or Unclear. Methods are anonymized (as A or B) and questions for each explanation type are randomized. Ten questions (five from both the EAT and HUNT policies) are asked for each explanation type giving a total of twenty questions to each participant. In addition, at the end of the study, we ask users to rate each explanation type based on a 5-point Likert scale for four qualitative metrics - completeness, sufficiency, satisfaction and understandability - as has been done in previous studies on explainable RL (Madumal et al., 2020). For users to familiarize themselves with the two types of explanations we also provided them with two training examples, one for each type at the start of the survey.

To be fair to VIPER-D explanations, rather than just displaying rules in text which may not be aesthetically pleasing, we also created a visualization which not only displayed the (five) rules to the user, but also three boards, one each for pacman, the ghost, and the pill, highlighting their possible locations as output by the rule. This is beyond the typical decision tree offered by VIPER, which is meant for continuous state spaces, and better renders what the explanation looks like in our discrete setting. Screenshots of sample visualizations along with the instruction page and optional feedback left by users can be found in the appendix.

The study was implemented using Google Forms and we received 37 responses from people with quantitative/technical backgrounds, but not necessarily AI experts. We removed 5 responses as they were likely due to users pressing the submit button multiple times as we twice received multiple answers within 30 seconds that were identical.

**Observations:** Figure 4 (left) displays user accuracy on the task for method SSX and Viper-D. Users clearly were able to better distinguish between the EAT and HUNT policies given explanations from SSX rather than Viper-D and the difference in percentage correct is statistically significant (paired t-test p-value is 0.01). Another interesting note is that less than 5% of SSX explanations were found to be Unclear whereas more than 25% of Viper-D explanations were labeled Unclear, meaning that, right or wrong, users felt more comfortable that they could extract information from SSX explanations.

Figure 4 (right) displays the results of the qualitative questions ("Was it complete/sufficient/satisfactory/easy to understand?") for both SSX and Viper-D which users rate on a 5-point scale ranging from "Not at all" to "Yes absolutely". All metrics score high for SSX and differences with Viper-D are statistically significant. These results are consistent with the very different percentage of Unclear selections for SSX and Viper-D, i.e., users found very few SSX explanations to be unclear and therefore also scored SSX higher in the qualitative metrics.

## 7 DISCUSSION

We have seen in this work that our novel approach of identifying strategic states leads to more complete, satisfying and understandable explanations, while also conveying enough information needed to perform well on a task. Moreover, it applies to single agent as well as multi-agent adversarial games with large state spaces. Further insight could be distilled from our strategic states by taking the difference between the variables in some particular state and the corresponding strategic state and conveying cumulative actions an agent should take to reach those strategic states (viz. go 2 steps up and 3 steps right to reach a door in Four Rooms). This would cover some information conveyed by the typical action-based explanations we have seen while possibly enjoying benefits of both perspectives. Other future directions include experimenting to see if strategic states could be used as intermediate goals for efficient training of new policies and extension of our idea to continuous state spaces. While one could discretize the state space which could be suboptimal, it would be interesting to see if it can be symbiotically done.

ETHICS STATEMENT

As Deep Reinforcement Learning (DRL) has seen huge success in solving challenging problems with superhuman performances across multiple domains (Silver et al., 2016; 2018; Mnih et al., 2015) understanding these policies has implications in human learning as well as safety (Garcia & Fernandez, 2015). We in this work have provided a mechanism for uncovering the core insights from such policies. Saying that, though, it is possible that some insights might be missed by our method as it distills relevant information. This limitation is not specific to our method and applies to other posthoc explanation methods as well. From a privacy standpoint our method could be used to decipher sensitive information about an agent. Mitigation may be possible by restricting access of our method to only desirable parts of the state space that are deemed safe to divulge.

REPRODUCIBILITY DISCUSSION

All datasets used are public. Experimental details are in Section 5 of the main paper and Section C in the appendix. Code is provided in the supplement.

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

## APPENDIX

## A  ALGORITHMIC DETAILS

We first prove Proposition 1.

*Proof.* Consider two sets $U$ and $V$ consisting of strategic states of meta-state $\Phi$, where $U \subseteq V$. Let $w$ be a strategic state $\notin V$ and $G_\Phi(.)$ represent the objective in equation 3, then we have

$$G_\Phi(U \cup w) - G_\Phi(U) = C(w, \Phi) - \lambda \sum_{u \in U} \max\left(\gamma(w, u), \gamma(u, w)\right) \tag{4}$$

Similarly,

$$G_\Phi(V \cup w) - G_\Phi(V) = C(w, \Phi) - \lambda \sum_{v \in V} \max\left(\gamma(v, w), \gamma(w, v)\right) \tag{5}$$

Subtracting equation (5) from (4) we get,

$$(4) - (5) = \lambda \left( \sum_{v \in V} \max\left(\gamma(v,w), \gamma(w,v)\right) - \sum_{u \in U} \max\left(\gamma(w,u), \gamma(u,w)\right) \right) \qquad (6)$$

$$= \lambda \sum_{v \in V \setminus U} \max\left(\gamma(v,w), \gamma(w,v)\right) \geq 0 \qquad (7)$$

Thus, the function $G_\Phi(.)$ has diminishing returns property. $\qquad \square$

We next comment on the convergence for Algorithm 1 which follows directly since our objective in equation (2) is bounded and monotonically decreases at each iteration.

**Proposition 2.** *Meta-state finding Algorithm 1 converges.*

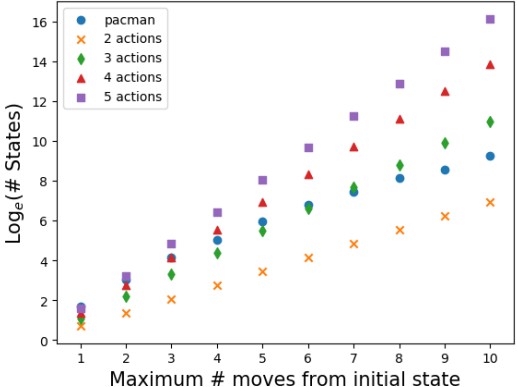

Figure 5: Illustrations of SSX state space size in minipacman. Worst case state space size for local approximations is $N^M$ where $N$ is the maximum number of moves made and $M$ is the number of possible actions per move. Pacman's state space is averaged over 100 random samples for each $N = 1, \ldots, 10$. The state space of minipacman, while also growing exponentially, grows much slower (like a game with 2-3 actions per move) which makes SSX a practical method for such games.

## B    ADDITIONAL PRACTICAL CONSIDERATIONS

**Additional information on scalability:** SSX is applied in Section 5 to games with state spaces ranging from small to exponential in size. The SSX algorithm is straightforward for small state spaces as one can pass the full state space as input, however, neither finding meta-states nor strategic states would be tractable with an exponential state space. One approach could be to compress the state space using VAEs as in (Abel et al., 2019), but as shown in Figure 1c, interpretability of the state space can be lost as there is little control as to how states are grouped. The same phenomenon can be observed when considering compression versus explainability in other contexts such as classification models. Our approach is to use local approximations to the state space; given a starting position, SSX approximates the state space by the set of states within some $N > 0$ number of moves from the starting position. Considering different starting positions will offer the user a global explanation for a fixed policy. In this approach, Algorithms 1 and 2 are a function of $N$, i.e., increasing $N$ increases the size of the approximate state space which is passed to both algorithms. One can contrast our approach of locally approximating the state space with that of VIPER (Bastani et al., 2018) which uses full sample paths to train decision trees.

Figure 5 displays how the state space size in minipacman, discussed in Section 5, grows in practice as the number of possible moves $N$ allowed for the local approximation grows. Worst case state space size for local approximations is $M^N$ where $M$ is the number of possible actions per move. At any position on the board, minipacman has at most 4 possible actions (3 possible directions to move or stay) and the ghost has an additional 3 potential actions for a total of 7 possible state movements at most. The state space of minipacman is averaged over 100 random samples for each $N = 1, \ldots, 10$

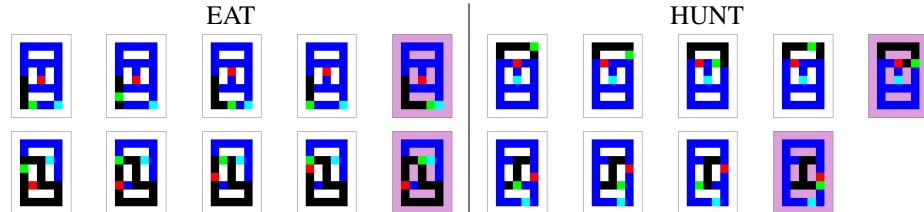

Figure 6: Illustration of selecting important states on minipacman. Two policies, EAT and HUNT, are displayed across two scenarios for each. For each scenario, a single cluster is shown. For a given cluster, the last board with pink background is an important state for that cluster as defined by equation (D). The color scheme is as follows: green = pacman, red = ghost, yellow = edible ghost, cyan = pill, blue = food, black = food eaten, white/pink=wall.

and, while growing exponentially, acts similar to a game with between 2-3 actions per move because most states in the local approximation are duplicates due to both minipacman and the ghost going back and forth. When enumerating the local state space, duplicates can be removed before increasing the length of possible trajectories so that the local state space stored does not grow at the maximum rate in practice. Also note that the size of the local approximation to the state space will not be affected if the board size increases because only local states are considered.

**Storing Paths:** The predecessor matrix $P$ is defined such that $P_{ij}$ is the predecessor of state $j$ on a shortest path from $i$ to $j$ (and infinity if no such path exists). This matrix is used to retrieve the shortest path between any two states $i$ and $j$. Then a strategic state is defined as a state $s'$ such that $P_{st} = s'$ where $\phi(s) = \phi(s') \neq \phi(t)$, i.e. $s'$ is the last node on the shortest path between states $s$ and $t$ that are in two different meta-states that lies in the same meta-state as $s$. Then, by this definition, we can penalize the number of strategic states.

**Number of Meta-states** $k$**:** The number of meta-states can be chosen using standard techniques as trying different $k$ and finding the knee of the objective (i.e. where the objective has little improvement) or based on domain knowledge. State representations may affect the (appropriate) number.

## C  REPRODUCING THE EXPERIMENTS

Code for SSX experiments is included. Separate directories and corresponding README files are available for experiments pertaining to the three domains: Four Rooms, Door-Key, and Minipacman. Training of all models uses default parameters from the respective github repositories used for each environment (links in Section 5). Parameters used for the experiments are given in Table 1.

Table 1: Parameters used for Four Rooms, Door-Key, and Minipacman experiments

| Parameter | Domain | | |
|---|---|---|---|
| | Four Doors | Door-Key | Minipacman |
| # strategic states $k$ | 2 | 5 | 5 |
| $\lambda$ from eq. (3) | 50.0 | 1.0 | 0.1 |
| $\epsilon_g$ from algorithm 2 | 0.1 | 0.1 | 0.1 |
| $N$ # steps used for local approximation to $\mathcal{S}$ | NA | 6 | 6 |

## D  COMMENTS ON STATE IMPORTANCE

A potential baseline for explanations is motivated by the policy summarization method Amir & Amir (2018). Explanations in that work are offered as simulated trajectories that are deemed important. Importance of states along the trajectory is defined by

$$I(s) = \max_a Q^{\pi_E}_{(s,a)} - \min_a Q^{\pi_E}_{(s,a)},$$

| Minipacman | | | | | Ghost | | | | | Food | | | |
|---|---|---|---|---|---|---|---|---|---|---|---|---|---|
| $N$ | 3 | 4 | 5 | 6 | $N$ | 3 | 4 | 5 | 6 | $N$ | 3 | 4 | 5 | 6 |
| 3 | 0 | 0.70 | 0.92 | 1.14 | 3 | 0 | 0.88 | 1.05 | 2.59 | 3 | 0 | 0.52 | 0.73 | 0.84 |
| 4 | - | 0 | 0.85 | 0.78 | 4 | - | 0 | 1.36 | 2.23 | 4 | - | 0 | 0.57 | 0.93 |
| 5 | - | - | 0 | 1.29 | 5 | - | - | 0 | 1.73 | 5 | - | - | 0 | 0.88 |

Table 2: Faithfulness measures for size of local neighborhood. For Minipacman, measures are distances of minipacman positions in strategic states when using different local approximations with varying approximation size $N$, where $N$ is the maximum number of steps allowed for a state to be included. Distances are symmetric (hence use of -).

where $Q_{(s,a)}^{\pi_E}$ is the Q-value for policy $\pi_E$ for taking action $a$ at state $s$. By definition, this gives a measure of variation (by action) for a given state, i.e., the large the potential impact of which action is taken, the higher the importance value. While this measure can successfully be used to select important trajectories that give users an idea of what a policy is doing, as done in (Amir & Amir, 2018), such important states are not necessarily good representatives of states that one should aim for, as is the goal of strategic states in SSX. Figure 6 depicts a few examples to demonstrate this property on minipacman. Note that there is a negative reward for eating a pill in the EAT scenario because the policy is to eat all the regular food while avoiding the ghost. Q-values were estimated using the equation, Q(s,a) = r(s,a) + E[V(t)], where $r(s, a)$ is the reward for taking action $a$ in state $s$ and is determined by the scenario (EAT or HUNT), $t$ is the state that pacman transitions to by taking taking action $a$ in state $s$, and $V(t)$ is the value of being in state $t$. The expectation is taken with respect to the new state (because the ghost moves stochastically after pacman takes action $a$).

In both EAT scenarios, important states are found with pacman located directly next to the food; indeed, these are important according to the measure in equation (D) because the difference of whether pacman moves away from the pill or eats it is large. However, these are not positions where one would expect to guide pacman in the EAT scenario. Similarly, for the HUNT scenarios, the important states are found where pacman is relatively close to the ghost. For example, in the first row, if pacman moves left, there is a chance the ghost will move right and eat pacman, whereas if pacman moves up or down, pacman will continue.

## E  FAITHFULNESS AND CONSISTENCY OF LOCAL APPROXIMATIONS

We here investigate how sensitive strategic states in SSX are to the size of the local approximation. Let $N$ be the maximum number of steps that can be taken (as referred to above in the discussion on Scalability in Section 3.4). We allow $N$ to vary from 3 to 6 steps and run SSX for the minipacman setup from various starting boards along a trajectory of the HUNT scenario. For each trial, we take the priority strategic state of the cluster containing the initial board and compare it against trials from the same initial board but with different $N$. We consider three distance metrics for each comparison: the distance between minipacman positions, the distance between ghost positions, and the distance between remaining food indicators of the boards.

Table 2 displays the results. Distance is measured using $l_2$ norm on the difference of (x,y) coordinates for minipacman and the ghost and the difference of indicators of whether or not food is present for the food. For reference, a value of 1 for minipacman means that minipacman was usually 1 position away (horizontally or vertically) in the two strategic states being compared. Values close to 0 mean that minipacman was often in the same spot in respective strategic states. We first note that distance generally increases the larger the difference of $N$ between trials, as expected. The second note is that the distance of the ghost is larger than that of minipacman, which is also expected as the ghost moves randomly whereas minipacman is driven by the policy.

We next consider the consistency of local approximations by comparing strategic state results from an initial state to that when slightly perturbing the initial state. Consistency is illustrated on minipacman, where perturbations are done by randomly removing 3 pieces of food from the initial state being considered by SSX. For each initial state, 10 additional random perturbations are used, and various initial states along a trajectory are used. Distances are measured asThanks described above for minipacman, the ghost, and the food. Results are given in Table 3. Again, minipacman on average

|  | Avg Distance |
| --- | --- |
| Minipacman | 0.63 |
| Ghost | 1.61 |
| Food | 1.32 |

Table 3: Consistency measures for perturbations of the initial state. For Minipacman, measures are distances of minipacman positions in strategic states when using different initial states (comparing a strategic state from initial state to that of a perturbed initial state).

is less than one position away when comparing SSX results between two initial states that differ by a minor perturbation. As expected, the ghost moves more since the ghost is not controlled by the policy. The distance of 1.32 for food corresponds to a difference of close to 3 pieces of food (since distance is an $l_2$ norm between the indicator matrices of food). This makes sense as the perturbations make the boards differ by 3 units of food.

## F    ADDITIONAL MINIPACMAN EXAMPLES

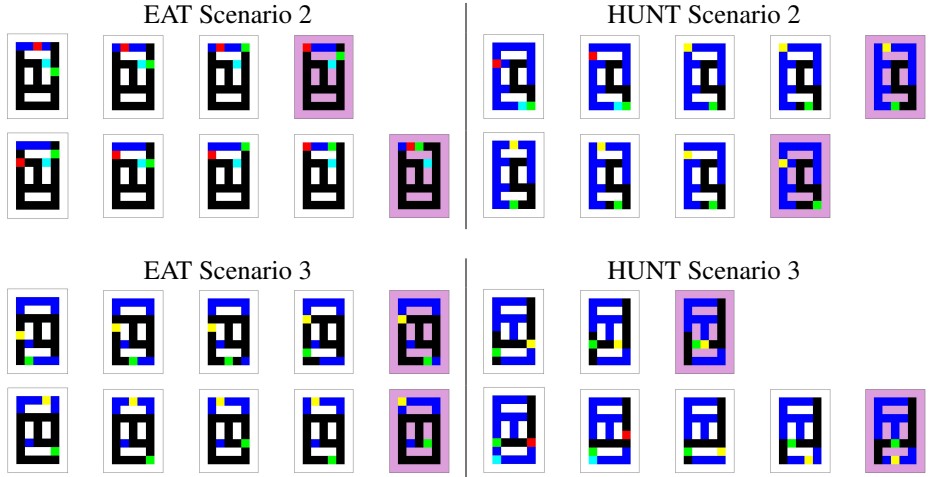

Figure 7: Illustration of our SSX method on two additional scenarios of minipacman. Two policies, EAT and HUNT, are displayed across the two scenarios. For each scenario, two clusters, one per row, are shown as part of the SSX result. The last board with pink background is a strategic state for each cluster. The color scheme is as follows: green = pacman, red = ghost, yellow = edible ghost, cyan = pill, blue = food, black = food eaten, white/pink=wall. In EAT scenarios, pacman generally ignores the pill and stays away from the ghost (even if the pill has been eaten). In HUNT, pacman generally looks for the pill (but stays away if the ghost is near it) and moves toward the ghost (if the pill has been eaten).

Figure 7 shows results of SSX applied to two additional scenarios for the minipacman example. EAT Scenario 2 shows pacman willing to take a chance of being eaten in order to get more food and EAT Scenario 3 shows that, even though pacman already ate the pill (the ghost is yellow when the pill is eaten), pacman prefers to eat more food rather than head for the ghost. These strategic states contrast directly with those in the HUNT scenarios. Strategic states in Hunt Scenarios 2 and 3 also show pacman eating the pill in order to hunt the ghost rather than eating more food.

## G    ADDITIONAL USER STUDY INFORMATION

Participants in the user study first read a set of instructions where the two environments, hunt and eat, are explained. Further, participants are detailed the color scheme and what they are expected to do with each question of the survey. The full set of instructions are given in Figure 8. Participants

were then first trained on one example each with SSX and Viper-D explanations and were given the reasoning one might use in making their choices. The respective training examples are given in Figures 9 and 10. Following the training examples, participants went through a series of 20 questions, 10 from each of SSX and Viper-D explanations. The distribution of correct answers for eat versus hunt is 50/50. An example of each type of question with the choices, Eat, Hunt, and Unclear, are shown in Figures 11 and 12.

## AI Explanations for Pacman

PLEASE READ CAREFULLY

Thank you so much for volunteering to take this survey. No information other than what you choose to provide will be recorded in this survey. There are 22 questions and it should take around 10-15 minutes to complete the survey. At the end you can also leave additional feedback. Below are the details of the survey.

Two AI methods were trained on a simplified version of the popular game Pacman, where we have just one Ghost, one Pill and a smaller maze having slightly different objectives.

HUNT: The first method we call HUNT declares Pacman as a winner if he is able to eat the Ghost (after eating the Pill which looks like a big light blue square) without getting eaten himself. There is no reward for eating food as in regular Pacman.

EAT: The second method we call EAT declares Pacman as a winner if he is able to eat all the food (excluding the pill). Eating the ghost in this case does not buy him any extra browny points.

Each question in the survey asks you to guess which one of these two objectives are being explained by the provided visualizations. There are two different types of visualizations, or what we call explanations, which we refer to as Type A and Type B. Your task is to use the provided explanation (either of type A or type B) to surmise the objective (HUNT or EAT) of the underlying method. At the end, we will also ask which of these explanation types you prefer based on a 5 point Likert scale covering qualitative aspects such as completeness, sufficiency, satisfaction and understanding.

In all cases, the question is: WHICH METHOD (HUNT or EAT) IS THE EXPLANATION OF TYPE (A or B) EXPLAINING? You also have the option to mark UNCLEAR if you are unsure about the objective given the visualization.

Note the following color scheme for the different entities in the game.

Color scheme applicable to both explanation types:
Pacman: Green
Pill: Light blue
Ghost before pill is eaten: Red (Ghost hunting Pacman)
Ghost after pill is eaten: Yellow (Ghost can be eaten by Pacman)
Location where wall is present: White (background)

There are also explanation specific colors (which will become clear in the next two examples) as follows:

Applicable to type A explanation:
Location where food is present: Dark blue (maze paths)
Location where food is eaten: Black (maze paths).
Target state for each row: Pink (background).

Applicable to type B explanation:
Same color scheme as indicated in the common part. However, since these are rule based explanations regions in the maze are highlighted corresponding to the rule, rather than just specific locations of the different entities (viz. Pacman, Ghost, etc.) in a particular state.

We now provide one example of each explanation type before starting the survey. Thanks again for taking the survey.

* Required

Figure 8: Screenshot of user study instructions. SSX explanations are anonymized as Type A explanations and Viper-D explanations are anonymized as Type B explanations.

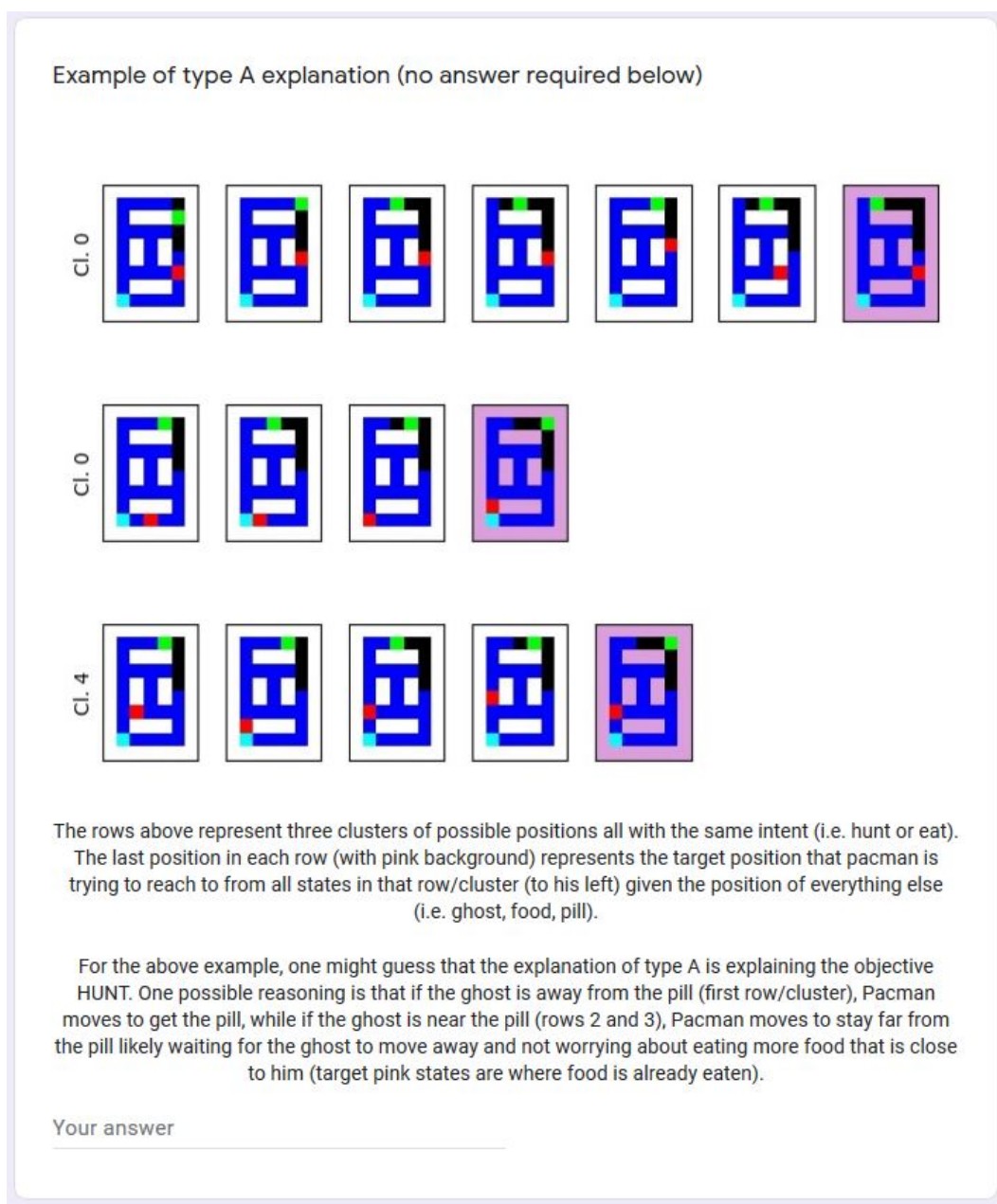

Figure 9: Screenshot of SSX explanation example used to train the participant taken from user study. SSX explanations are anonymized as Type A explanations.

Example of type B explanation (no answer required below)

The rows above represent five individual rules that each predict the same intent (i.e. hunt or eat). The title above each rule says what direction the rule says Pacman should move, along with additional details about how much food should be remaining and where food is around Pacman. The three boards in each row shows where the rule says Pacman, the Ghost, and the pill should be located (locations being denoted by the green, red, and light blue squares, correspondingly). For example, if Pacman, the ghost, and the pill are in the potential locations dictated by the board, and food is to the right and below Pacman, the rules says that Pacman should move down.

For the above example, one might guess that the explanation of type B is explaining the objective EAT. One possible reasoning is that the rules suggest Pacman move away from the ghost but, not necessarily, towards, the pill. For example, the last rule suggest Pacman move to the right because there is no food to the left, but if Pacman were hunting, he'd probably want to go left and get to the possible pill locations faster.

Your answer

_______________________________

Figure 10: Screenshot of Viper-D explanation example used to train the participant taken from user study. Viper-D explanations are anonymized as Type B explanations.

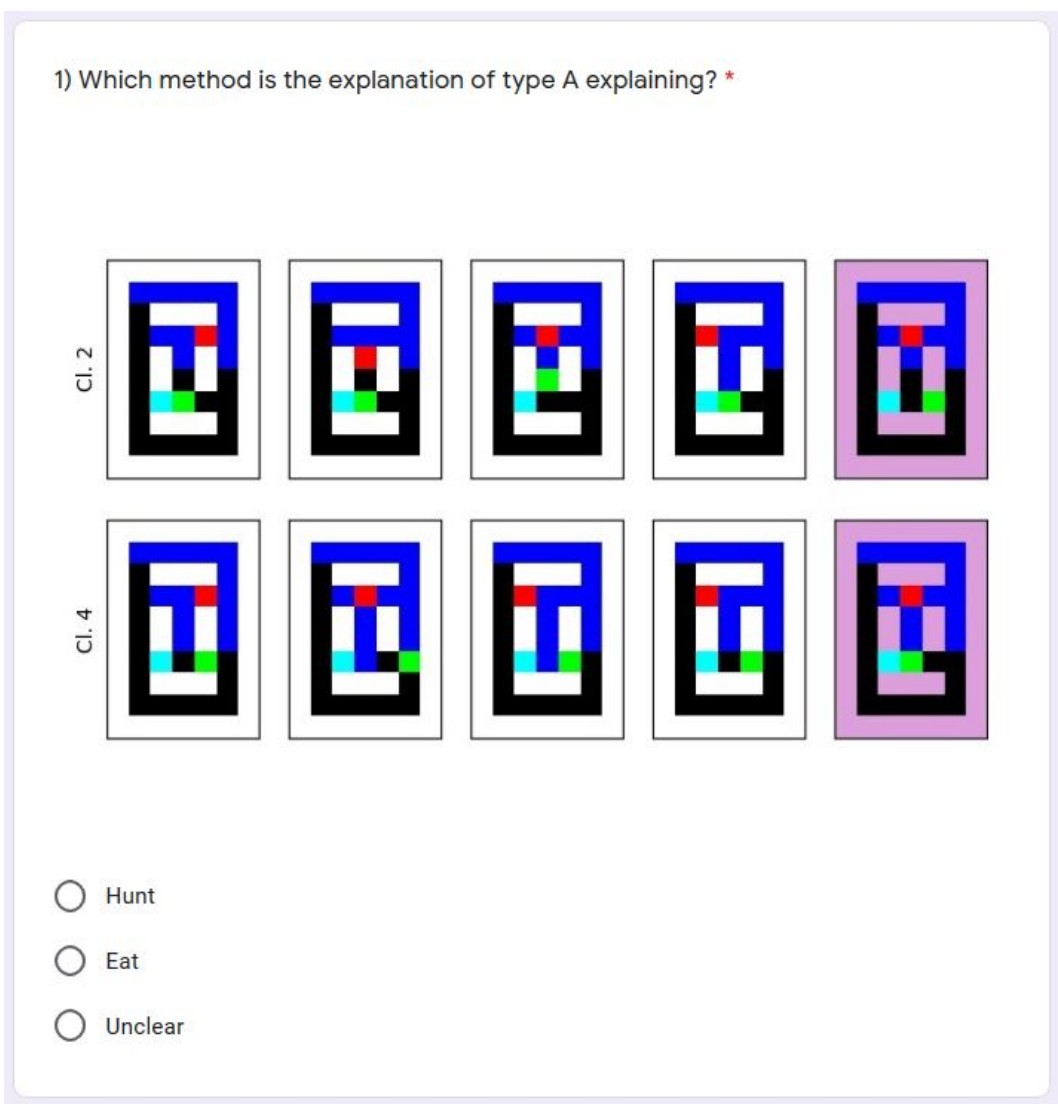

Figure 11: Screenshot of SSX explanation survey question taken from user study. SSX explanations are anonymized as Type A explanations.

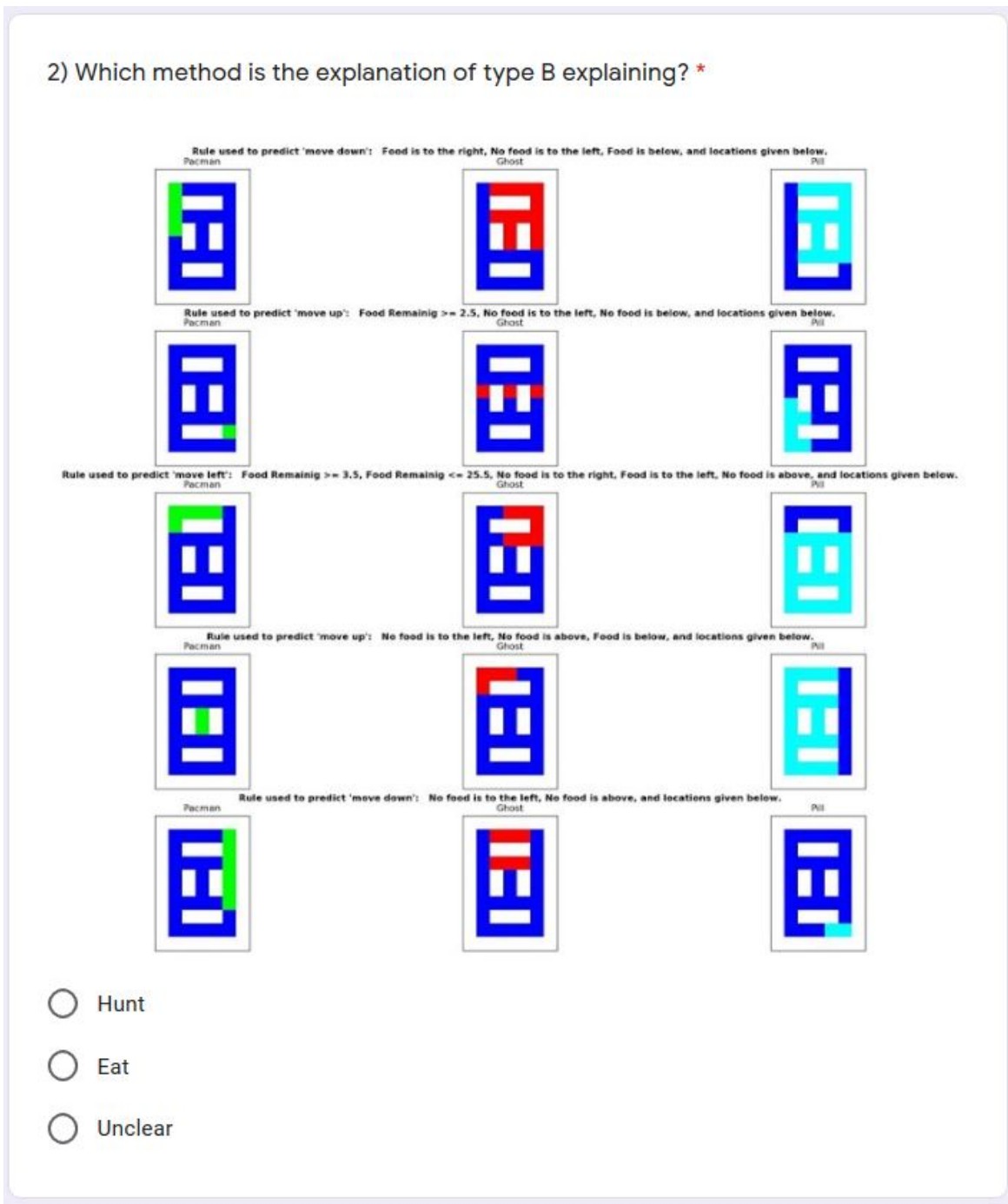

Figure 12: Screenshot of Viper-D explanation survey question taken from user study. Viper-D explanations are anonymized as Type B explanations.

