# OpenReview forum: "Interpreting Reinforcement Policies through Local Behaviors"
_ICLR.cc/2022/Conference — ICLR 2022 Submitted_

### Official Review · Reviewer_53LC · 2021-10-26

**Correctness:** 3
**Technical Novelty And Significance:** 2
**Empirical Novelty And Significance:** 2
**Recommendation:** 5
**Confidence:** 3

**Details Of Ethics Concerns:**

The paper provides a discussion on ethical concerns.

**Main Review:**

This paper explains DRL policies from an interesting perspective, i.e., explaining via prediction. It draws insights into a DRL agent's policy by predicting certain steps of future moves. Although studying an important problem, this paper has the following limitations that prevent me from supporting it.

1. The readability and clarity of this work can be further improved. Due to some unclear terminologies, the proposed technique is not clearly described. Specifically, I would first suggest giving a concrete example to demonstrate the difference among states, meta states, and strategy states, especially strategy states. Neither Fig. 1 nor notions clearly express that. In addition, the papers suddenly throw out many termonologies without explanations, such as state-space topology, bottleneck, strategic policy. To be best of my knowledge, these terminologies are not specific to RL and may cause confusion. I would suggest giving a clear explanation or definition before using them.

2. The proposed technique is actually mainly algorithms 1 and 2, where Algorithm 1 is like a clustering algorithm, and Algorithm 2 is a simple searching algorithm. I thus am a little bit concerned about the technical contribution of this work. It may not be reaching the bar of a top-tier ML conference. Regarding Algorithm 1, I am also wondering if one can directly apply an existing clustering method to replace it, like using K-NN or K-means based on $\Gamma$.

3. The paper misses a potential baseline that also explains DRL policies through action prediction, i.e., [1]. I would suggest the authors discuss the difference between [1] and the proposed technique and even empirically compare them.  Regarding the user study, existing works utilize user study to compare the utility difference among different explanation methods in identifying good/bad policies, e.g., [2,3]. I would suggest the authors comment on their choice of user study design and why not follow the existing work to compare policy identification ability.

4. It's not clear the problem space of the proposed technique or, in other words, what types of RL tasks are suitable for the proposed technique. More specifically, I am wondering whether the proposed technique can be applied to widely used Atari and MuJoCo environments or even real-world environments, like Go and Texas hold 'em.

[1] Explain via prediction: What Did You Think Would Happen? Explaining Agent Behaviour through Intended Outcomes, NeurIPS 20.

[2] Visualizing and understanding atari agents, ICML 2018.

[3] Highlights: Summarizing agent behavior to people. AAMAS 2018

**Summary Of The Paper:**

This paper proposes a novel explanation method for DRL policies that clusters environment states into meta states and identifies strategic states from meta states as explanations. The paper empirically compares its proposed explanation method with some existing methods on three games and designs a user study to demonstrate its utility.

**Summary Of The Review:**

This paper studies an important problem from an interesting perspective. It explains a policy by predicting certain steps of future moves. In summary, I identify the following limitations from this work:

1. The readability and clarity of this work can be further improved. Due to some unclear terminologies, the proposed technique is not clearly described.

2. The technical contribution may be a little bit thin, and some design choices are not clearly justified.

3. The paper misses a potential baseline that also explains DRL policies through action prediction.

4. It's not clear the problem space of the proposed technique or, in other words, what types of RL tasks are suitable for the proposed technique.

---

> ### Author Response · Authors · 2021-11-16
> **Response to reviewer 53LC**
>
> Thank you for your review and detailed comments. We have taken your comments into account which has made the paper clearer, as well as addressed your other concerns below.
>
> **Re. Give clear definition/explanation of terminologies, such as states, meta states, and strategy states; state-space topology, bottleneck, strategy policy:** A new paragraph three has been added in the Introduction to clearly explain the terms state, meta-state, strategic state, and topology in reference to Figure 1. The term policy dynamics is explained at the end of paragraph four of the Introduction. We have not added more explanation for the term bottlenecks, as our only point is to distinguish them from explainability techniques.
>
> **Re. technical contributions of algorithm 1 and 2. What will be the result if directly apply existing clustering method, such as k-NN or Kmeans to replace algorithm 1:** Algorithm 1 is a novel extension of spectral clustering that regularizes by common states along high-probability paths. This regularization also allows one to consider specific cases such as $\eta\rightarrow 0$. In this case, the meta-states will tend to be equisized where the likelihood of transition between meta-states will be minimized leading to (approximate) optimization of an NCut objective (Luxburg, A Tutorial of Spectral Clustering, Statistics and Computing, 2007). For larger $\eta$, the likelihood of transition between meta-states is still kept small (which is desirable), but there is more of a tendency towards having few large clusters as we try to assimilate states that are on many paths from a meta-state. Such behavior cannot be guaranteed for other clustering methods such as K-Means. Algorithm 2 is, indeed, a search algorithm, but the objective is novel, where we prove it to be submodular making it possible to have efficient greedy solutions with constant factor approximation guarantees. Moreover, integrating these two techniques for interpreting RL policies ---to our knowledge--- is nontrivial and has not been explored in a principled way (proving convergence and sub-modularity, and providing user study) before. Notably, many well known techniques such as DQN were built on old known techniques, such as fitted q-learning, replay buffer, target network, which might not be novel from RL’s perspective, yet integrating them into DQN in a principled way allowed for significant advancements in deep RL research.
>
> **Re. baseline that also explains DRL policies, including Explain via prediction, Visualizing and understanding atari agents, HIGHLIGHTS:** Thank you for bringing our attention to [1] and [2], which have been noted in the Related Works section. [1] is a local explanation method but should be viewed similarly to self-explaining models because the explanations are learned during training (see eq (7) and (8) of [1]). This provides useful and interesting explanations, but unlike SSX, requires access to the training algorithm of the policy being explained. Regarding user studies, our task-based study is in similar spirit to the user studies done in [2] as well as HIGHLIGHTS. In [2], the task is to identify the important features in the images that are used by an agent, and the task is applied to the policy with and without the explanation. In HIGHLIGHTS, the task is to decide among two policies which performs better, and they compare HIGHLIGHTS trajectory explanations against two baselines, the first $k$ trajectories and $k$ randomly sampled trajectories, which clearly HIGHLIGHTS should outperform. Our user study task is similar and also asks a user to decide among two policies, but to label them EAT vs HUNT rather than good vs bad; good vs bad is another option but in some respects would be easier to distinguish. Our user study is more in line with other user studies in explainability (e.g., ``Leveraging Latent Features for Local Explanations" by Luss, et.al. in KDD 2021) in that we compare against another established work that is separate from ours making it a much stronger baseline than what is done in HIGHLIGHTS. We believe this to be noteworthy as we could not ourselves find another RL explainability paper which performs a user study comparing its own proposal to a well established prior method.

---

> > ### Author Response · Authors · 2021-11-16
> > **Response to reviewer 53LC continued...**
> >
> > **Re. What types of RL tasks are suitable for the proposed technique:** Our method can be applied to tasks that have long-term goals that require fulfilling different subtasks to complete this goal. This includes tasks that benefit from state abstractions (e.g., meta-states in SSX) and temporal abstractions (e.g, strategic states in SSX), and specifically to explain non-myopic policies.
> >
> > Towards this end, we have also given thought to where our method can be applied beyond grid games. We believe our perspective can be highly useful in domains such as robotics as well as business applications that leverage RL. For example, in robotics consider the task of lifting up a cup on a table using a robotic arm. Although simple for a human there are multiple subgoals a machine has to achieve in order to complete the task. It first needs to reach a state where the arm is lifted followed by a state where it is straightened and moved over the table. It then has to be lowered towards the cup, followed by the palm opening with fingers outwards and then closing the fingers around the cup to grasp it. Finally, the arm has to be lifted to pick up the cup. This whole sequence represents an MDP where the indicated important states would be strategic states for the task of lifting a cup across different initial states and setups. The meta-states would be groups of states indicating the various resting positions of the arm initially or states where the arm is raised or where it is stretched out to different degrees. Many such applications of our methodology could be envisioned for different tasks in the robotics space.
> >
> > For business applications, our approach has value as well. Consider scenarios where businesses want to increase their loyalty base. In such situations companies many times train RL policies that recommend to them the next based actions in terms of promotions and advertisements that they could offer/show the user based on their profile. Thus, they try to maximize the *Lifetime Value (LTV)* (Theocharous et al. Personalized Ad Recommendation Systems for Life-Time Value Optimization with Guarantees, IJCAI, 2015) as opposed to some myopic metric such as click through rate (CTR). For such (recommendation) policies our SSX method could be used to identify strategic offers that lead to many user upgrades and becoming loyalty customers based on a state space determined by user characteristics such as demographics, buying behavior etc. Thus, our approach is quite applicable in practice.

---

> ### Author Response · Authors · 2021-11-30
> **Checking in**
>
> Dear Reviewer 53LC,
>
> We hope that you find our responses below to be helpful. In particular, our revision has added much text to better explain all the terminology and we've given thought as to how SSX can be applied to real-world applications in robotics and business. With the additional experiments and revisions, we hope you find the paper improved. As the window for discussion is coming to an end, please let us know if we can further clarify anything.
>
> Thank you,
>
> Authors

---

> > ### Comment · Reviewer_53LC · 2021-12-03
> > **Responding to the authors**
> >
> > Thanks the authors for your rebuttal. I have read them and appreciated your efforts in clarifying the terminologies and differences with other literature. I have posted my new thoughts in the reviewer discussion (blind from authors). IMHO, I do not think other clarifications w.r.t. my reviews are needed at this stage.

---

> > > ### Author Response · Authors · 2021-12-03
> > > **Thank you for the response**
> > >
> > > Thank you for responding. In any case, we hope we have increased your confidence in our paper, and if more clarifications are required at any stage, we would be happy to provide them.

---

### Official Review · Reviewer_u9SV · 2021-11-01

**Correctness:** 3
**Technical Novelty And Significance:** 3
**Empirical Novelty And Significance:** 3
**Recommendation:** 6
**Confidence:** 4

**Main Review:**

As motivated by the first paragraph of the paper, interpreting RL policies such that a human user understands the reasoning behind these policies is a critical step toward using these policies in a trustworthy manner in real-world domains. Unfortunately, the paper does not show the usefulness of the proposed method on any real-world application. Studied applications are grid games that may not reflect real-world applications such as those that arise in queuing, dynamic pricing, revenue management, among many others. As explained above, the notion of meta-states and strategic states for applications beyond gird games is unclear. Particularly, how should a human user interpret strategic states in real-world applications? The interpretation of strategic states for these applications may not be as simple as doors in a grid game. Therefore, at the very least, I recommend authors provide detailed explanations on how their approach can be employed in real-world settings, i.e., in robotics and business applications. Moreover, conducting a numerical study that applies the proposed method to a real-world application can significantly add to the paper's contributions.

The authors assume that the dynamics of the expert policy π_E, denoted f_E, are known apriori, and thus Γ can be computed exactly and provided to Algorithm 1 as an input. However, such dynamics are often unknown in practice and should be estimated through policy simulation, that is, one can only estimate f_E by simulation of π_E. Authors need to pose their method and the computation of matrix Γ through policy simulation.

To build on my previous concern, computing the entire matrix Γ, as done in the current version, is expensive and seems unnecessary. In many domains, a good policy (i.e., the expert policy) only visits a small subset of states with a non-negligible probability. In this case, is not it enough to restrict the computation of Γ to such states? If so, authors can more efficiently compute Γ by only focusing on the states observed by π_E during simulation. Please also comment on the computational cost of getting the eigen decomposition of the Laplacian of Γ needed in Algorithm 1. Can this be efficiently computed for large-scale problems? Given the current manuscript, I am not convinced that the computation of Γ and its Laplacian eigen decomposition are tractable for large-scale problems.

The authors mentioned the convergence of their method multiple times in the paper without carefully explaining what their algorithm converges to.  Proposition 1 proves the convergence of Algorithm 1 without identifying what does Algorithm 1 converges to. Can authors show that if a given expert policy induces a collection of k meta-states \{Φ_1^E,…,Φ_k^E \}, then Algorithm 1 converges to this set of meta-states? If not, how the output of Algorithm 1 is connected to the expert policy? The current version of Proposition 1 is not insightful. Moreover, the convergence results in Algorithm 1 should be presented when Γ is estimated through policy simulation and not when Γ  is known exactly (please see my comments above).

I wonder how the choice of parameter η affects the interpretability of the expert policy. Some theoretical or empirical studies are useful here. For example, the authors can explain in the extreme cases of η being zero and being a large number, how does Figure 1-a change? Is there a need for fine-tuning η to get human-understandable meta-states? Moreover, how does Figure 1-a change for different values of λ?

It is not clear if the program (3) has a unique solution or not. I think there are examples at which multiple strategic states exist, given a fixed m and meta-state. In the case of degeneracy, how should a human user choose a solution? I encourage authors to comment on this.


Minor comments.

Consider rewriting the sentence “… which act as intermediate goals for states belonging to a particular meta-state”. I had read this sentence multiple times to understand what a strategic state means.

Term “policy dynamics” is unclear in Section 1. Authors may need to explain what they mean by policy dynamics in that section.

Coloring in Figure 1 is hard-to-follow. Please consider using different colors and using different markers.

The sentence “While one could discretize the state space, it would be interesting to see if it can be symbiotically done” is misleading. In many real-world continuous-state MDPs, discretizing the state-space leads to poor approximations and is thus not a solution.

Figure 2 is very confusing. I had to read the text and check the figure multiple times. Is there a way to enhance it?

**Summary Of The Paper:**

The paper proposes an approach to interpret a black-box control policy of a reinforcement learning (RL) problem such that its interpretations can be understood by a human user. For a given expert policy (i.e., a fitted deep neural network), the proposed approach uses transition probabilities induced by the expert policy to define regions of the state space that are “similar.” These policy-dependent regions are referred to as meta-states and are computed using an algorithm similar to spectral clustering. The proposed method then computes so-called strategic states for each meta-state, where strategic states of a meta-state are those states that belong to the meta-state and bridge the meta-state to other ones. In other words, the expert policy should pass through the strategic states frequently when it goes from one meta-state to another. The paper performs numerical experiments using standard RL applications, i.e., four rooms, door-key, and mini-Pacman, to show the effectiveness of the proposed approach.

**Summary Of The Review:**

My overall impression of the paper is positive, but I have several concerns. My most critical concern is that given the theory and numerical experiments in the current manuscript, I am not sure if the proposed approach delivers human-understandable interpretations of RL policies in applications beyond grid games arising in robotics, flight control, multi-armed bandit, etc. In other words, the proposed method seems to work only for the grid games, and it is unclear how a human user should interpret meta-states and/or strategic states in applications beyond grid games.

---

> ### Author Response · Authors · 2021-11-16
> **Response to reviewer u9SV**
>
> Thank you for the positive feedback and constructive criticism. We address your comments below.
>
> **Re. Applicability beyond grid games:** We believe our perspective can be highly useful in domains such as robotics as well as business applications that leverage RL. For example, in robotics consider the task of lifting up a cup on a table using a robotic arm. Although simple for a human there are multiple subgoals a machine has to achieve in order to complete the task. It first needs to reach a state where the arm is lifted followed by a state where it is straightened and moved over the table. It then has to be lowered towards the cup, followed by the palm opening with fingers outwards and then closing the fingers around the cup to grasp it. Finally, the arm has to be lifted to pick up the cup. This whole sequence represents an MDP where the indicated important states would be strategic states for the task of lifting a cup across different initial states and setups. The meta-states would be groups of states indicating the various resting positions of the arm initially or states where the arm is raised or where it is stretched out to different degrees. Many such applications of our methodology could be envisioned for different tasks in the robotics space.
>
> For business applications, our approach has value as well. Consider scenarios where businesses want to increase their loyalty base. In such situations companies many times train RL policies that recommend to them the next based actions in terms of promotions and advertisements that they could offer/show the user based on their profile. Thus, they try to maximize the *Lifetime Value (LTV)* (Theocharous et al. Personalized Ad Recommendation Systems for Life-Time Value Optimization with Guarantees, IJCAI, 2015) as opposed to some myopic metric such as click through rate (CTR). For such (recommendation) policies, our SSX method could be used to identify strategic offers that lead to many user upgrades and becoming loyalty customers based on a state space determined by user characteristics such as demographics, buying behavior, etc. Thus, our approach is quite applicable in practice. It is also worth noting that a common theme, as seen above, for applicability in the real-world is to explain policies that are not myopic.
>
> **Re. estimating $\Gamma$:** We have now mentioned in the paper that $\Gamma$ can be estimated by simulating the policy at the end of ``Maximum likelihood (expert) paths" subsection of Section 3.
>
> **Re. Tractability of $\Gamma$ and eigen decomposition of the Laplacian:** Yes you are correct that $\Gamma$ can be restricted only to regions of the state space that are more likely, which will result in improving efficiency. Regarding eigen decompositions, the python function scipy.sparse.linalg.eigsh is "a wrapper to the ARPACK [1] SSEUPD and DSEUPD functions which use the Implicitly Restarted Lanczos Method to find the eigenvalues and eigenvectors." These rely on Fortran77 subroutines and the use of these ARPACK functions is the classic method for large-scale eigenvalue problems. Another good library is PRIMME (https://github.com/primme/primme). Two important notes about the computation: 1) SSX only needs $k$ eigenvectors where $k$ is the number of clusters desired which is typically small, and 2) the Laplacian of $\Gamma$ is typically highly sparse which makes it amenable to algorithms tailored for sparse matrices like the one we use. For sparse matrices, the algorithm upon which ARPACK is based has complexity $O(nk^2)$ where $n$ is the matrix size and $k$ is the number of eigenvectors desired (``Dynamic Thick Restarting of the Davidson, and the Implicitly Restarted Arnoldi Methods" by Stathopoulos, Saad, and Wu, SIAM Journal on Scientific Computing, 1997).

---

> > ### Author Response · Authors · 2021-11-16
> > **Response to reviewer u9SV continued...**
> >
> > **Re. Algorithm 1:** The output of Algorithm 1 is connected to the expert policy because the $\Gamma$ matrix is determined by the expert policy. As stated in the "Maximum likelihood (expert) paths" subsection in Section 3, ``The distance we consider is the most likely path from state $s$ to state $s'$ under $\pi_E$." These distances form $\Gamma$. Regarding Proposition 1, convergence is the strongest statement we can hope to obtain for such a problem in general, where the final solutions may be local optimas or saddle points, etc. However, for specific cases such as $\eta\rightarrow 0$ the meta-states will tend to be equisized where the likelihood of transition between meta-states will be minimized leading to (approximate) optimization of an NCut objective (Luxburg, A Tutorial of Spectral Clustering, Statistics and Computing, 2007). For larger $\eta$, the likelihood of transition between meta-states is still kept small (which is desirable), but there is more of a tendency towards having few large clusters as we try to assimilate states that are on many paths from a meta-state. Regardless, we have moved Proposition 1 to the Appendix Section A (now called Proposition 2) to make room for more important details.
> >
> > **Re. Effect $\eta$ and $\lambda$ on Figure 1a:**
> > Higher values of $\eta$ will result in less equi-sized clusters as it will bias states to be added to a cluster that has many paths linking to it. For reasonable values of $\eta\in (0,5]$ we found our algorithm to be quite stable where the meta-states were almost unchanged w.r.t. those observed in Figure 1a. $\lambda$ controls diversity of the strategic states. However, in Figure 1a the most important (or first) strategic state selected for each of the meta-states (i.e. the bigger markers) will not change irrespective of the value of $\lambda$ as it will have effect only following the first selection. If $\lambda$ is close to 0 we will end up picking strategic states that are close to the first selected strategic state since, states around it are likely to also have many crossing paths. However, choosing reasonable sized $\lambda$ (i.e. $\in (0.1,5]$) will result in picking diverse strategic states that also lie on a reasonable number of crossing paths as seen in Figure 1a.
> >
> > **Re. Solution to Eq. 3:** We employ a deterministic greedy algorithm which has a constant factor approximation guarantee (Krause, 2021) as our objective is submodular. Solutions will typically be unique as it is unlikely to get the exact same value for the objective in Eq. 3 for two different strategic states, although not provably so. Ties could be broken randomly. In Figure 1a, for example, the two strategic states for the green meta-state, the two larger green circles, should be equally good as they are equidistant from the goal state, but nonetheless the one indicated by the bigger circle was preferred indicating that ties are unlikely.
> >
> > **Re. rewriting the sentence “… which act as intermediate goals for states belonging to a particular meta-state”:** We have rewritten the sentence to make it clearer to readers: ``Our approach involves two steps: 1) learning meta-states, i.e., clusters of states, based on the dynamics of the policy being explained, and 2) within eat meta-state, identifying states that act intermediate goals, which we refer to as *strategic states*."
> >
> > **Re. explain meaning of policy dynamics:** We have added the following explanation: ``We use the term *policy dynamics* to refer to state transitions and high probability paths. We use the term dynamics because this notion contrasts other methods that use actions to explain what to do in a state or to identify important states; strategic states are selected according to the trajectories that lead to them, and these trajectories are implicitly determined by the policy." It appears in Section 1 after other methods are discussed because it is easiest for a reader to understand in relation to these other methods.
> >
> > **Re. Coloring in Figure 1 is hard-to-follow:** We have updated Figure 1 to use different colors and markers, and used different sized markers to differentiate strategic states, with the larger marker denoting the priority strategic state in a meta-state. The caption has been updated to reflect the changes.

---

> > > ### Author Response · Authors · 2021-11-16
> > > **Response to reviewer u9SV continued...**
> > >
> > > **Re. The sentence “While one could discretize the state space, it would be interesting to see if it can be symbiotically done” is misleading:** We have now rephrased the sentence to indicate that discretization of the state space could be a sub-optimal solution. We would like to note, however, that an explanation, in order to be consumable and not overwhelming, in some sense has to be a small set of finite concepts (Miller, Explanation in Artificial Intelligence:
> > > Insights from the Social Sciences, Artificial Intelligence 2019) such as our strategic states.
> > >
> > > **Re. Figure 2 is very confusing:** Thank you for pointing this out. We agree the images in this figure could use additional text for the reader, and have added the following further explanation in the second paragraph of the Door-Key subsection of Section 5: ``The scenes in Figure 2 are exactly what a user sees. To better understand why the scenes do not appear easily connected, consider the first two states in the first row - the only difference from the first state is that the agent has changed directions. When facing the wall, the agent's view only includes the three positions to the right and one position to the left. All positions on the other side of the wall are not visible to the agent, which is depicted as black. When the agent changed directions (row 1, column 2), many more positions in the room become visible to the agent."

---

> ### Author Response · Authors · 2021-11-30
> **Checking in**
>
> Dear Reviewer u9SV,
>
> We hope that you find our responses below to be helpful, in particular, to illustrate how SSX can be useful in real-world applications in robotics and business. We have also provided more details as to the scalability and computational aspects regarding the $\Gamma$ matrix that you inquired about, along with answers to your other questions, and modified the paper to address your suggestions. With the additional experiments and rewriting to clarify terms, we hope you find the paper improved. As the window for discussion is coming to an end, please let us know if we can further clarify anything.
>
> Thank you,
>
> Authors

---

> ### Author Response · Authors · 2021-12-06
> **Any more clarifications?**
>
> Dear Reviewer u9SV,
>
> You had a positive impression of our paper before the rebuttal, and we would very much like to hear if our responses further improved your view of our work. We understand that you are likely very busy, but please find a moment to verify if our responses and changes to the paper address your concerns.
>
> Much appreciated,
>
> Authors

---

### Official Review · Reviewer_GrkK · 2021-11-02

**Correctness:** 2
**Technical Novelty And Significance:** 2
**Empirical Novelty And Significance:** 3
**Recommendation:** 3
**Confidence:** 5

**Main Review:**

#### Local vs Global Confusion:

-The authors state "our focus is on local explanations," and use this to discount existing "local" works while not comparing to "global" works. However, in other places in the text, SSX is treated as producing global explanations. SSX is "local" only in the sense that, for larger domains, approximations are made to address scalability issues. Especially when discussing experiments, the authors use explanations to comment on overall (global) policy behavior (e.g., study subjects are asked about overall behavior not per-state behavior).

-As a specific example, TLdR automatically finds landmark states based on transitions between states. This approach does differ from SSX but not to the degree mentioned in this work (which labels it a "global summary" method, unlike SSX). At the very least, TLdR is a good candidate for comparison.

#### Complexity Concerns:

-The authors note that directly applying SSX is not feasible, so a local neighborhood is used. The faithfulness of such a local approximation is not evaluated (e.g., run approximation for different N and non-approximate approach, then compare resulting explanations).

-However, each neighborhood requires a number of samples exponential in the "radius" of the neighborhood. Experiments use a "radius" of at most 6. This is low for the use-cases used to motivate this work. To produce explanations for larger domains, even a single local neighborhood may not be feasible to explain.

-The authors do not present a way to combine local neighborhoods. As neighborhood "radius" is decreased (to address exponential complexity) the number of neighborhoods to combine increases. It is unclear how to explain behaviors that consist of more actions than the neighborhood diameter. The consistency of explanations for adjacent explanations is not evaluated (e.g., produce Figure 1 for different neighborhoods, then look at whether states labeled by multiple neighborhoods create a consistent explanation).

-The sharding process implicitly relies on access to a simulator to efficiently gather samples centered around different states. This is not acknowledged as a requirement/shortcoming of SSX.

#### Positioning wrt Related Work:

-Most relevant past work has been mentioned! Unfortunately, mentioning this work is insufficient to position SSX with respect to past work. SSX's methodology is indeed different from past work, but its final product (set of clusters with representative example(s) for each) is not novel. The authors note that some past work has "the intent of efficient learning rather than interpretability" but a different intent does not make a method sufficiently novel. Please position SSX with respect to past global methods, making the differences clear.

-SSX is only compared with VIPER-D. Other works should be included in this comparison, especially works that produce similar cluster / meaningful state explanations. Relatedly, "they require estimating Q-values ... which relies on sensitive hyperparameter tuning" does not seem like a valid reason to exclude a work from this comparison.

-The authors note that other state clustering methods may not map adjacent states to the same meta-state. However, SSX does not do always do this either, as seen in Figure 2.

#### Problems with Example Explanations:

-Explanations for Fig 2-3 are based on starting states within a single trajectory. I am concerned that the clusters exhibit "common" behavior within themselves specifically because the starting states are spaced apart. (e.g., if neighborhoods have radius 6 and clusters are m>6 steps apart, then of course each cluster will have similar states) To test this, multiple trials should be run and similarity should be measured across trials. Is SSX effectively just segmenting a trajectory for these figures? Please explicitly test this by comparing to baseline methods (such as creating a single "cluster" from each neighborhood around each starting state).

-There appears to be manual tuning that is insufficiently explained. How were the starting states chosen for the different neighborhoods? How were clusters chosen for showcasing? (i.e., Fig 3 has odd cluster numbers, as though some were manually chosen to show)

-The explanations given for Fig 2 (arguing that the clusters exhibit meaningful, distinct behaviors) do not match what is shown in the figure. There are images in later clusters despite occurring earlier in the episode. Specifically, "Unlocked Door, second row, third column" is several steps through the door despite being part of the "going through the door" cluster, despite "third row, second column" being an earlier state and in the "moving toward the target" cluster.

#### Problems with Human Study:

-It is great that a human study was done. The right feedback is being gathered from human subjects.

-VIPER-D and SSX differ among several axes, so SSX's success cannot be attributed to the specific qualities pointed out by the authors. For example, VIPER-D uses a different output representation (distinct from both SSX and original VIPER).

-In addition to not including past works in this comparison, a number of baselines are missing to determine which aspects of SSX are helpful. Please add these comparisons to show that SSX is meaningfully better due to its clustering approach. Perhaps showing states leads to subject satisfaction (should compare to randomly grouped past experiences)? Perhaps the trajectory is being segmented by clusters (should compare to "local neighborhood around equally spaced starting states")?

-VIPER-D is unnecessarily modified and likely performing worse than original VIPER. VIPER can already handle discrete states. VIPER is designed to work with an iterative state gathering method, not simply the full state space as done here.

**Summary Of The Paper:**

This work proposes to explain an RL policy by clustering states into meta-states and presenting strategic state(s) for each meta-state. This clustering is performed based on policy rollouts, balancing likelihood of paths within a meta-state and number of paths from states within the meta-state. The authors present example explanations for three domains. A user study is performed to compare VIPER-D to the proposed method (SSX).

**Summary Of The Review:**

The authors present a new way (SSX) to create a certain style of explanation, but the style itself is not novel. SSX is not shown to be better than existing methods. Furthermore, its computational complexity is concerning, and the authors do not fully address how to apply their method to larger domains.

---

> ### Author Response · Authors · 2021-11-16
> **Response to reviewer GrkK**
>
> Thank you for the detailed feedback. We hope the following responses address your concerns.
>
> **Re. local vs global confusion:** We do not mean to discount local explanations. By local, we mean that SSX offers a user an explanation for what a policy is doing at a current position. The explanation explicitly focuses on the local neighborhood and implicitly does not account for what will happen too far into the future. This should be viewed as complementary to other local explanation methods which offer different intuitions, e.g., contrastive explanation methods or those that explain a specific action. Global methods are by nature summarizations and should also be used as a complementary tool by users. It is also true that SSX can provide global intuitions in certain situations. The Four Rooms game in Figure 1 has a small state space that does not require any local approximation and hence SSX is able to provide a global view, i.e., since the state space is small, the local strategy is equivalent the global strategy. In the case of minipacman, we consider the policy globally stable in that the intent of EAT or HUNT does not change among different localities. Since the global intent coincides with what the policy does locally, we are able to use it in this manner in the User Study, and furthermore, since most baselines, e.g., VIPER, are global methods, it makes sense to compare them in this setup.
>
> **Re. Comparisons:** TLdR, as written, is considered a global explanation, though, in theory, we agree that it could be applied to local neighborhoods. However, this work also relies on expertise in planning and using a specific planning software (FastDownward from https://github.com/aibasel/downward) which is not easily adaptable without the right knowledge making it out of the scope of this paper. Moreover, the focus is on plans for stochastic shortest path problems. Furthermore, as explained in the TLdR paper, conversion of problems in our settings require adding new actions to the model (see Section 5.2 of TLdR paper).
>
> Nevertheless, another comment of yours below about Q-Values made us look back at the HIGHLIGHTS paper.
> Although not directly applicable, given the actor-critic implementation of minipacman, we have now estimated Q-Values using Q(s(t),a(t)) = r(s(t),a(t)) + E[V(s(t+1)] and applied this measure in our SSX framework in Appendix Section D, which also shows why such importance measures do not capture the insight provided by strategic states.
>
> **Re. Faithfulness of local neighborhood:** We have conducted an experiment that varies the local approximation to use a local state space that varies from 3 to 6 steps out for minipacman and added it in Appendix Section E. Faithfulness of the approximation is given by a distance measure between the top learned strategic states of the clusters in which the initial state is located. We measure 3 distances: distance between pacman in the strategic states, between the ghosts in the strategic states, and the difference in food between the strategic states. As expected, the distance increases slightly the farther apart the approximations, and the ghost positions differ the most because the ghost move randomly. This shows that our method is faithful.

---

> > ### Author Response · Authors · 2021-11-16
> > **Response to GrkK continued...**
> >
> > **Re. Larger domains:** As illustrated, SSX already works on useful games and generates meaningful explanations. Regarding larger domains, SSX can still be applied in a similar manner with the understanding that the explanation concerns what to do in the next X steps. In theory, the state space of larger domains can also be further discretized and SSX can applied to a discretized version of the larger state space. For example, works on compression/abstraction of policies could potentially be used and SSX then applied on top of the abstraction. However, unlike the abstraction methods known to the authors, one would need an abstraction where the abstract states are themselves interpretable in order to obtain a meaningful explanation from SSX. While a few points were made in the Scalability subsection for Section 3, we also refer you to the Appendix Section B which gives much additional details on scalability as well as details on experiments that further illustrate the scalability of SSX (please see Figure 5 in the Appendix). The insightful section was held to the Appendix in the initial submission due to space limitations. Another scaling issue for SSX, also noted by Reviewer u9SV below, involves the eigen decomposition of the Laplacian of the $\Gamma$ matrix, which is scalable. We utilize the scipy.sparse.linalg.eigsh function which is ``a wrapper to the ARPACK [1] SSEUPD and DSEUPD functions which use the Implicitly Restarted Lanczos Method to find the eigenvalues and eigenvectors. This remains the standard state-of-the-art for such decompositions and is extremely fast especially when one desires only a handful of eigen vectors. It is important to note that 1) SSX only needs $k$ eigenvectors where $k$ is the number of clusters desired which is typically small, and 2) the Laplacian of $\Gamma$ is typically highly sparse which makes it amenable to algorithms tailored for sparse matrices like the one we use.
> >
> > **Re. combining neighborhoods:** As a local method, we have not yet considered how to combine local neighborhoods, though we agree that this is an interesting direction to think about. Moreover, considering the faithfulness experiments described above, and consistency experiments we describe next, it is worth noting that SSX is quite stable relative to locality size and hence aggregation may not be required.
> >
> > **Re. Consistency of explanations:** We have conducted a similar experiment as that done above for faithfulness and put the results also in Appendix Section E. In this experiment, we compare the priority strategic state SSX learns from an initial state with the priority strategic state SSX learns from random nearby initial states (we take the initial state and randomly take away food). Three distance metrics are measured the same as described above for faithfulness.
> >
> > **Re. Sharding Process:** We have added comments about the simulator requirement at the end of the the ``Maximum likelihood (expert) paths" subsection in Section 3, as computation of our $\Gamma$ matrix is the first place where the policy simulator is required in SSX. It is important to also consider that other explanation methods also require access to a simulator, e.g., HIGHLIGHTS must simulate trajectories because trajectories are essentially the output explanation and VIPER must simulate trajectories to generate a decision tree.
> >
> > **Re. Positioning wrt Related Work:** We are glad to hear that we covered the relevant past work sufficiently. Our use of the phrase ``more so with the intent of efficient learning rather than interpretability" was ill-worded and is rephrased. Such abstraction works do not offer explainability (see Figure 1c). State abstraction in the sense of these works is specifically to compress the state space in order to offer users a simpler policy that is learned on the compressed state space. Our point is that while there are such works on state abstraction, they should not be confused (or compared) with works on explainability that might also use the word "abstraction" as does TLdR. Another relevant note about TLdR to consider is that while, landmark states are output, there is no output of clusters, but rather a graph of how landmark states are related, which is quite different from SSX.

---

> > > ### Author Response · Authors · 2021-11-16
> > > **Response to GrkK continued...**
> > >
> > > **Re. Comparison with VIPER-D**: We note that to the best of our knowledge we know of no other related work that compares to prior state-of-the-art explainability works as we have done. HIGHLIGHTS is, in fact, the only paper with baselines, but these are just simplified versions of the HIGHLIGHTS method. Nevertheless, as mentioned above, we have implemented a new baseline motivated by the importance values in the HIGHLIGHTS paper, and have shown why this baseline does not offer explainability in our framework, but rather shows success when used as done with trajectories in the HIGHLIGHTS paper. As we also note below, we implemented VIPER ourselves as the original package was not easily adaptable due to very little comments, and we even had memory issues on the demonstration game they provided.
> > >
> > > **Re. Mapping adjacent states to the same meta-state:** We are not sure where we commented on this aspect in the paper. Could you please provide more details?
> > >
> > > **Re. Fig 2-3 are based on starting states within a single trajectory:** We believe there is a misunderstanding with what is clustered. Explanations are not based on a single trajectory. Rather, first the local neighborhood is determined from the initial state based on the maximum number of steps (e.g., 6). This is the only place where the number of steps comes in. Then clustering of all those states is done based on all possible trajectories within the neighborhood. Strategic states are then determined for each cluster but very close states can be in different clusters.
> > >
> > > **Re. Manual tuning:** Initial states were selected from random trajectories that were simulated. For the visuals in the paper, we used random trajectories that are included in the submitted code package under directory "saved\textunderscore board" in the corresponding folders for minipacman and door-key. From these random trajectories, we selected diverse clusters to showcase the utility of the method and the varied insights it offers in different settings. This is a standard way of showing qualitative examples in prior works on explainability as well, e.g., TlDR, and in more general explainability methods, e.g., LIME ("Why Should I Trust You?" by Riberio, Singh, Guestrin in KDD 2016) or CEM (``Explanations based on the Missing: Towards Contrastive Explanations with Pertinent Negatives" by Dhurandhar, et.al. in NeurIPS 2018). Note that HIGHLIGHTS and VIPER do not even show qualitative examples in the paper (though HIGHLIGHTS does provide a link to a single video of trajectories). Fig 2 and 3 and corresponding text has been modified so the reader is not confused by different cluster numbers and is explained that each row is a different cluster from the SSX explanation.
> > >
> > > **Re. Explanations for Fig 2 do not match the figure:** Indeed, in the Unlocked Door, second row, all states have the agent in the same room as the reward. We now call the the strategic state ``made it through the door" to be more specific. It is not yet moving toward the reward as the agent does in the third row where we see examples of the agent moving closer. Regarding the comparison between the second and third row, note that the third column in the second and third row for Unlocked door are the same state. The second row explains a starting state 6 steps into the trajectory and the third row explains a starting state 9 steps into the trajectory. As these are explaining different starting points, it is possible that they have overlapping clusters as occurs here, and this is further illustrates why we consider each explanation to be local. In minipacman this can also happen because minipacman moves back and forth over positions already covered. We have added a paragraph at the end of the Door-Key subsection in Section 5 to address this issue.

---

> > > > ### Author Response · Authors · 2021-11-16
> > > > **Response to GrkK continued...**
> > > >
> > > > **Re. Problems with Human Study:** We are glad you agree the correct feedback is being gathered. It is true that VIPER-D and SSX differ among several axes, and this is true for other methods, e.g., HIGHLIGHTS, as well. However, the actual output of VIPER-D is a tree representation that would not be interpretable with our feature space. Hence, we created a much more appealing and visually intuitive representation of the output of VIPER-D (see Figure 9 in the Appendix of the updated submission - note that it is also in the original submission). Our representation visually shows what part of the feature space the rules given by the tree apply to, which is much easier to a user than seeing numerical rules for positions of pacman, ghost, and pill in the game. We believe this is the best one can do to make the comparison between the two methods fair. Further, we again state that this is the only explainability paper in this domain that conducts a user study against another explainability method. HIGHLIGHTS, the only other paper with a user study, compares against trajectories that can be considered simplistic versions of the method that it would clearly outperform.
> > > >
> > > > **Re. VIPER-D being modified:** While there is a package for VIPER, it is not easily adaptable for other users (lack of comments). We even had a difficult time getting the example case (pong) to run (Out of Memory on our GPUs) and could not adapt it to our method. As is typical for experiments in many cases, we had to re-implement a version ourselves.

---

> > ### Comment · Reviewer_GrkK · 2021-11-19
> > **Reply to authors**
> >
> > Some of my concerns have been addressed, but most remain, so I cannot recommend that this work be accepted in its current state.
> >
> > Selecting starting states from a single trajectory: My concern is that several starting states are sampled along a single trajectory. The local neighborhoods are then gathered around these states (for a sequence of "local" explanations). When the agent performs a sequence of different behaviors, these behaviors would be split between these starting states rather than via the clustering process itself.
> >
> > Manual tuning: "we selected diverse clusters to showcase the utility of the method and the varied insights it offers in different settings" is problematic. Hand-picked examples do not accurately reflect the method's ability to explain agent behavior. Past work relying on qualitative examples does not justify the heavy reliance on them in this work. Criticizing VIPER for "not even show[ing] qualitative examples" is odd-- VIPER specifically took the correct approach of performing quantitative evaluation.
> >
> > Explanations for Fig 2 do not match the figure: We are in agreement, then, that the presented explanations do not allow a user to draw the longer-horizon conclusions desired by the authors. The distinction between different stages of solving the task is specifically because of initial state selection.
> >
> > Comparison with VIPER-D: Past work performing poor comparisons / lacking comparisons is not reason to follow suit. Furthermore, this work is not the only one that performs comparisons to past methods-- consider looking at other works that cite VIPER to see examples. Note that I am pushing for a proper quantitative comparison, which need not be through a user study.
> >
> > Problems with Human Study: SSX consists of several components, and these should be ablated (as noted in my original review). This has not been addressed. To demonstrate that SSX's partitioning of agent behavior is not due to selection of initial states for neighborhood creation, alternate methods of forming the clusters (one per neighborhood, random, etc) should be compared.
> >
> > Larger Domains: The proposed approaches do not address the concerns I raised about SSX's computational complexity.
> >
> > Faithfulness+consistency of local neighborhood: Thank you for performing these experiments! The results are promising.
> >
> > VIPER-D being modified: I am concerned that a re-implementation was performed, but my specific concern is that VIPER has been *modified* in ways that unnecessarily depart from the original VIPER work. On the tangential topic of re-implementation: I have personally used and extended the VIPER codebase. Likewise, subsequent similar works (e.g., MoET) successfully use unmodified VIPER. The inability to run VIPER on Pong due to low GPU memory is not relevant-- this is not a shortcoming of VIPER nor the available VIPER implementation (the original NN expert is what uses GPU memory; VIPER uses no GPU memory).

---

> > > ### Author Response · Authors · 2021-11-20
> > > **Thank you for responding, but you misunderstand local explanations (in XAI), the value of showing qualitative examples, which quantitative comparisons are possible between SSX and trees, and our scalability arguments [1\3]**
> > >
> > > Thank you for taking the time to read our responses. We are glad that you found our new experiments that you suggested to be promising. We understand that concerns remain and we further address them below:
> > >
> > > **Re. Selecting starting states from a single trajectory:** We believe there is a misunderstanding in what we deem a local explanation. SSX offers an explanation at a specific initial state. Given a single initial state, SSX gathers only the local neighborhood around this state, performs clustering of these states, and selects strategic states among these clusters. Each row of the figures explains the policy at a particular position (i.e., runs the SSX algorithm from scratch at a position). The SSX explanations at each row are independent of one another (and that is how it is also possible for there to be an overlap of states in the explanations, as explained in the additional text that has been added in the second paragraph of Door-Key subsection starting with ``To better understand why the scenes do not appear easily connected...". One of our main contributions is this general approach to explainability for RL: clustering a local neighborhood to form meta-states and finding the strategic states within the resulting meta-states. In practice, as one asks for explanations along a trajectory, the local neighborhood of future explanations can thus overlap.
> > >
> > > **Re. Manual Tuning:** A good evaluation in explainable AI (XAI) should include both qualitative and quantitative evaluations. Furthermore, simulating the behavior of a black-box model with a human who consumes explanations is a preferred way of evaluating XAI methods (Lipton, 2016; Doshi-Velez and Kim, 2017). Please consider two important points about evaluations in XAI:
> > >
> > > 1) Quantitative evaluations generally illustrate properties of the explanation, e.g., VIPER shows how many nodes are required to achieve certain rewards versus another decision tree. Such quantitative evaluations are also only effective in comparing against very similar explanations, e.g., VIPER compares against DAGGER, which is also a tree but not actually a tree used for explanations. There are no quantitative evaluations against other methods outside of trees as it is not clear how to conduct such an evaluation.
> > >
> > > 2) The most important aspect of an explanation is whether or not a human user can understand it [Miller, Explanation in Artificial Intelligence: Insights from the Social Sciences, Artificial Intelligence 2019], as without that, the explanation will not be used no matter the quantitative evaluation. VIPER has no such qualitative evaluations likely because the trees used for quantitative evaluations are not actually explainable to a human (e.g., verifying robustness of Pong extracted a tree with 769 nodes, and comparisons with DAGGER used trees ranging from 31 nodes to 769 nodes). The only visual in VIPER is for a toy pong model and used for verification, not explainability.
> > >
> > > **Re. Comparison with VIPER-D:** While VIPER has been cited many times, we have only been able to find a comparison to VIPER in the MoET paper (which you only made us aware of now) that you refer to in your last comment, and not in any other references that cite VIPER (to our knowledge). Please note that MoET is an unpublished work and according to the ICLR guidelines, authors "may be excused for not knowing about papers not published in peer-reviewed conference proceedings or journals, which includes papers exclusively available on arXiv." While you previously noted that ``Most relevant past work has been mentioned," we would be happy to consider other published works that we should discuss if you could point them out. Our next point expands on the quantitative evaluations we feel are best suited for SSX.
> > >
> > > **Re. Further points about quantitative evaluation:** Note that VIPER, MoET, and DAGGER are all decision trees and so metrics for accuracy and how many nodes they require to get the same accuracy can easily be done. However, quantitative metrics for comparing against other methods are not clear, i.e., how to compare VIPER against HIGHLIGHTS, TlDR, or SSX. Our quantitative metrics were initially restricted to scalability (via Figure 5 in the Appendix) and thanks to your feedback, we have added quantitative evaluations on faithfulness and consistency that you find promising. Note that neither VIPER, nor any other explainability works in RL to our knowledge, have such quantitative metrics.

---

> > > > ### Author Response · Authors · 2021-11-20
> > > > **Thank you for responding, but you misunderstand local explanations (in XAI), the value of showing qualitative examples, which quantitative comparisons are possible between SSX and trees, and our scalability arguments [2\3]**
> > > >
> > > > **Re. VIPER-D being modified:** There are several points worth mentioning about our VIPER-D implementation:
> > > >
> > > > 1) The overall algorithm of VIPER is easily implementable (e.g., sampling new trajectories, aggregating trajectories into the dataset, sampling from the dataset, training a decision trees, and finally outputting the best tree) so we feel comfortable that our comparison is valid.
> > > >
> > > > 2) VIPER claims to use the CART algorithm for training decision trees. We originally considered our implementation an adaptation because we implemented VIPER using sklearn.tree.DecisionTreeClassifier which it turns out (we admit we did not realize before) uses an optimised version of the CART algorithm. We now refer to VIPER-D simply as ``our implementation of VIPER." Regardless, other decision tree algorithms could be used in VIPER as well so this modification would be minor. We keep the D in VIPER-D to distinguish the output of our implementation which is possible due to the discrete nature of the state space, and still maintain that this output makes the VIPER explanation more visually understandable than a tree would be.
> > > >
> > > > 3) An important aspect left out of VIPER actually deals with explainability - there is a tradeoff that is ignored between interpretability of the tree and accuracy (which is how they determine the best policy through cross validation). In our comparisons, we have taken interpretability of the tree into account, as should be done when using VIPER for explainability.
> > > >
> > > > 4) Another important note is that it is not clear that MoET, which compares against VIPER, used the implementation of the VIPER authors. MoET only cites the VIPER paper (not the code package), never claims to use the VIPER code package, and refers to the VIPER algorithm as is written in the Appendix of MoET.
> > > >
> > > > 5) We agree that memory issues could have been due to the policy in the VIPER demo, but regardless, due to the lack of comments, we found it difficult to work with (further justified by the lack of published works that used it), and we feel that re-implementing it should be viewed as a positive aspect to our paper.
> > > >
> > > > **Problems with Human Study:** We agree that ablations in a user study would be nice to have, however a further user study is obviously not feasible at this time. Moreover, given more time, there are physical limits to how many participants with a proper background that one can find that are willing to spend their time, and one can only ask a select number of questions in a user study in order to make the study reasonable for the participants. Regarding ablations, we believe that we have prioritized the right aspect to consider in a user study, and as you previously wrote, ``The right feedback is being gathered from human subjects." Lastly, please consider that this is the only paper, among those cited, that compares against another established method in a user study.
> > > >
> > > > **Explanations for Fig 2 do not match the figure: We are in agreement...** We are **not** in agreement. Each row in Figure 2 is offering an explanation for a different starting point. The explanations should be viewed separately, as the user is not looking to draw a longer-horizon conclusion. We clearly discuss this in the Door-Key subsection with the paragraph beginning with "For intuition on how a human would use these explanations..." Rows are explained separately. For example, we state ``The cluster in row
> > > > 1 for the Unlocked Door shows that the policy at these states is to face the door. Perhaps facing the door within a certain distance is how the policy breaks down the ultimate strategy." This explanation demonstrates that the explanation is about one task that must be completed on the way to finding the final goal. We agree that the distinction between the rows is because the initial states are different states along the trajectory. That is the point of the explanations we see.

---

> > > > > ### Author Response · Authors · 2021-11-20
> > > > > **Thank you for responding, but you misunderstand local explanations (in XAI), the value of showing qualitative examples, which quantitative comparisons are possible between SSX and trees, and our scalability arguments [3\3]**
> > > > >
> > > > > **Re. Larger Domains and concerns about computational complexity:** Two main concerns were listed: 1) Concerns about the faithfulness of local approximations: We have conducted suggested experiments which you have deemed promising. 2) Concerns about the radius of the neighborhoods: We believe there is a misunderstanding and hope our answer above ``Re: Selecting starting states from a single trajectory" better explains the SSX perspective because combining local neighborhoods, while an interesting direction, is not currently an aspect of the SSX explanation. Regarding larger domains requiring a radius larger than 6, we have proposed in our response that compression techniques could be used to address such issues, but that would be a direction for future work. Further, please refer to Figure 5 in the Appendix where experiments show that SSX scales well to a radius of 10. Also consider that a small radius might be sufficient for *local* explanations even on larger domains. This paper should be viewed as a new perspective for explaining policies and we hope one that will be built on further. If there is another stated concern that we did not address regarding complexity, please let us know so that we may address it.

---

> ### Comment · Reviewer_GrkK · 2021-11-22
> **Post author discussion update**
>
> Key concerns have still not been addressed. After reading the author responses and the other reviews, I believe this work is not ready for publication in its current state.
>
> Follow-up to latest author comments:
>
> Most importantly, SSX creates an explanation (row in Fig 2) based on states around a chosen initial state, but no evaluation is performed to determine whether SSX is improving over naive alternatives (e.g., "present all states around this initial state"). Ablations in quantitative experiments (as part of human study or with proxy metrics) are necessary.
>
> Shortcomings of the human study cannot be dismissed because "a further user study is obviously not feasible at this time". The authors chose to compare to an alternative method that differs along many axes, so the benefits of SSX have not been demonstrated in this study.
>
> The authors (correctly) note that quantitative comparisons between SSX and VIPER with proxy metrics are difficult. This is exactly why comparing to VIPER is a strange choice. The authors should, at the very least, ablate SSX to demonstrate that the clustering is useful and, ideally, compare to other methods for identifying abstract states.
>
> Presenting hand-picked examples for "qualitative" evaluation is not reason to skimp on quantitative evaluation.

---

> > ### Author Response · Authors · 2021-11-22
> > **Thank you for the continued discussion, but our human study follows standard practice in XAI for comparing methods, and our new quantitative evaluations (much thanks to you) greatly strengthen the utility of SSX and should not be considered skimping on quantitative evaluations.**
> >
> > As you still maintain concerns, we offer the following further comments to address further misunderstandings:
> >
> > **Re. Ablations:** We do not dismiss the use of ablations, but we do strongly contend that the primary utility of a user study in a work introducing a new method is to compare it against established methods. Our task-based user study is a standard setup to compare XAI methods in a user study. In fact, this aspect distinguishes our user study from that done in HIGHLIGHTS (which is the only user study done in the known related literature) because HIGHLIGHTS compares against baseline methods that clearly are not expected to do better. Our user study compares with the established VIPER method, and makes it as fair comparison as possible with the visuals we have created for VIPER (rather than displaying a tree that a human will not be able to easily parse).
> >
> >  **Re. Our choice of an alternative method for comparison:** As we mentioned in a previous reply and still maintain, VIPER is the best option among the related literature to compare against. In the case of minipacman, both VIPER and SSX can be used to distinguish between EAT vs HUNT because the intent of EAT or HUNT does not change among different localities. Regarding other possible comparisons, TLdR, as already mentioned, relies on expertise in planning and using a specific planning software (FastDownward from https://github.com/aibasel/downward) which is not easily adaptable without the right knowledge making it out of the scope of this paper. Moreover, the focus is on plans for stochastic shortest path problems, and furthermore, as explained in the TLdR paper, conversion of problems in our settings require adding new actions to the model (see Section 5.2 of TLdR paper). HIGHLIGHTS displays an explanation requiring a video to show full trajectories and is not amenable to comparing with static explanations like SSX or VIPER in a user study (and further deciding EAT vs HUNT would be obvious if one watched a full trajectory). Per your suggestion, we have used the importance metric from HIGHLIGHTS to design another baseline for selecting important states within SSX clusters. As already mentioned, Section D in the Appendix and Figure 5 illustrate that states selected using this metric do not offer the proper insight as strategic states do.
> >
> > **Re. skimping on quantitative evaluation:** Our paper currently offers quantitative evaluations to justify the scaling of the SSX (Figure 5), the faithfulness of local neighborhood approximation (Table 2), and consistency of explanations regarding the initial state (Table 3), all of which you clearly view positively. We do not believe this to be considered skimping on quantitative evaluations.

---

### Official Review · Reviewer_3j9K · 2021-11-03

**Correctness:** 4
**Technical Novelty And Significance:** 2
**Empirical Novelty And Significance:** 2
**Recommendation:** 5
**Confidence:** 3

**Main Review:**

Strength: (1) The paper proposes a new explanation method against deep reinforcement learning
Weakness: (1) The utility of the explanation is vague and needs more details. (2) the evaluation is insufficient.

**Summary Of The Paper:**

The paper presents an explanation method for deep reinforcement learning.

**Summary Of The Review:**

It seems the method cannot handle exponentially large state spaces. While the authors discuss the scalability issue, it is still difficult for readers like me to understand how to a bounded but exponentially large space. I would like to the authors provide more details.

I would like to see the proposed method applied in more sophisticated games (e.g., Pong and Mujoco games) in which the states cannot be numerated.

The utility of the explanation is unclear. It would be very helpful if the authors could share their survey questions. That could better help us understand the utility of the explanation.

---

> ### Author Response · Authors · 2021-11-16
> **Response to reviewer 3j9K**
>
> Thank you very much for your review. We hope the following responses help alleviate your stated concerns.
>
> **Re. large state spaces:** It is important to consider that SSX offers an explanation given a current state, i.e., addresses what is the policy doing given a user is at some given state. We are not offering an explanation for the entire state space, and this perspective is what allows us to offer an explanation of our type. By considering a local state space only X steps into the future, we can, in this case, propose an explanation that examines the entire local space. This is motivated by the idea that users often want explanations about their current situation; this is akin to post-hoc local explainability tools in classification where model decisions are explained for individual instances, i.e., why did a model predict class X for some specific instance. While a few points were made in the Scalability subsection for Section 3, we also refer you to the Appendix Section B which gives much additional details on scalability as well as details on experiments that further illustrate the scalability of SSX (please see Figure 5 in the Appendix). The insightful section was held to the Appendix in the initial submission due to space limitations.
>
> A main computational bottleneck in SSX, as noted also by Reviewer u9SV below, is the eigen decomposition of the Laplacian of the $\Gamma$ matrix. We utilize the scipy.sparse.linalg.eigsh function which is ``a wrapper to the ARPACK [1] SSEUPD and DSEUPD functions which use the Implicitly Restarted Lanczos Method to find the eigenvalues and eigenvectors". This remains the standard state-of-the-art for such decompositions and is extremely fast especially when one desires only a handful of eigen vectors. It is important to note that 1) SSX only needs $k$ eigenvectors where $k$ is the number of clusters desired which is typically small, and 2) the Laplacian of $\Gamma$ is typically highly sparse which makes it amenable to algorithms tailored for sparse matrices like the one we use.
>
> **Re. explanations for games where states cannot be enumerated:** SSX requires enumeration of the states, but in theory, this could still be applied to games like Pong by discretizing the local state space around a fixed position. In practice, any explanation that is a function of some property of the trajectories (e.g. \# out-paths of a node in SSX or satisfying the order of formulas in TLdR) must enumerate over states in some manner. These explanations should be viewed as complementary to those where explanations are themselves simulated trajectories (e.g., HIGHLIGHTS) rather than an enumerated set of states. More fundamentally, in order for an explanation to be consumable and not overwhelming, it has to, in some sense, be a small set of finite concepts (Miller, Explanation in Artificial Intelligence:
> Insights from the Social Sciences, Artificial Intelligence 2019) such as our strategic states.
>
> **Re. survey questions:** The survey questions are given in the "Setup" subsection of Section 6. We restate it here: We use the minipacman framework with the EAT and HUNT policies trained for Section 5 and each question shows either an SSX explanation or Viper-D explanation and asks the user ``Which method is the explanation of type A (or B) explaining?" to which they must select from the choices Hunt, Eat, or Unclear. Furthermore, Figures 8 and 9 in the Appendix show screenshots of the survey instructions and Figures 10 and 11 show screenshots of survey questions. Note these figures appear in the initial submission under different numbers as well.

---

> ### Author Response · Authors · 2021-11-30
> **Checking in**
>
> Dear Reviewer 3j9K,
>
> We hope that you find our responses below to be helpful, in particular, to clear up any confusion as to how our explanations can be applied to large state spaces, why enumeration is used, and regarding the computational aspects. With the additional experiments and rewriting to clarify terms, we hope you find the paper improved. As the window for discussion is coming to an end, please let us know if we can further clarify anything.
>
> Thank you,
>
> Authors

---

### Author Response · Authors · 2021-11-16
**General Response**

We thank all the reviewers for the detailed responses. One of the main common concerns is about comparisons. We would like to point out that, among the related work that we've discussed in the paper, our paper is the only explainability paper in this domain that compares against another established explainability tool, and we hope this aspect is viewed as an important distinction when considering empirical studies. We have tried to address many of your comments leading to changes in the submission that we list here so that all reviewers are aware of the updates:

    1) Appendix Section D: This new section details an experiment for a new baseline using an importance metric from HIGHLIGHTS   and new Figure 6 illustrates why importance can be misleading.
    2) Appendix Section E: This new section details both faithfulness and consistency experiments, as recommended by reviewers.
    3) Comments on the requirement of policy simulator and need for the simulator when $\Gamma$ cannot be computed exactly at the end of ``Maximum likelihood (expert) paths" subsection of Section 3.
    4) End of second paragraph in Section 4 is rewritten to clear up the difference between explainability and what some works refer to as state abstraction.
    5) Two additional references noted by Reviewer 53LC are added in Section 4.
    6) Figure 1 has been updated with different markers to make it clearer for the reader and text modified appropriately.
    7) Figures 2 and 3 have been updated. Text has been added to clear up confusion as to what rows are. Additional text has been added in second paragraph of Door-Key subsection starting with ``To better understand why the scenes do not appear easily connected..." in order to better explain what the images show to the reader.
    8) Row 2 of Unlocked Door Scenario is now called ``made it through the door". New paragraph added at end of Door-Key subsection addresses why states can reappear in different rows in the figure.
    9) Rewrote sentence in second paragraph of Section 1 that started ``Our approach involves learning meta-states...".
    10) New paragraph three in Section 1 to explain clearly various terms: state, meta-state, strategic state, and topology in reference to Figure 1.
    11) Additional text to explain the term ``policy dynamics"  at end of parapraph four of Section 1.
    12) Comments to mention new experiments added in end of first paragraph to Section 5, and end of ``Minipacman" subsection of Section 5.
    13) Last sentence of Section 7 has been updated.
    14) We also refer all reviewers to Appendix Section B, which might have been missed, but addresses some questions on scalability.

In order to fit these changes into the paper and maintain the original 9 pages, we moved (or removed or reformatted) the following items to the Appendix:

    1) Proposition 1 (now called Proposition 2) is now in the Appendix Section A.
    2) Subsection ``Storing Paths" from 3.4 is now in Appendix Section B.
    3) Scenarios 2 and 3 of Figure 3 (minipacman example) and associated text are in Appendix Section F.
    4) Comments on bottlenecks removed from caption to Figure 1.
    5) Last two sentences of caption to Figure 3 have been removed.
    6) Discussion has been made into one paragraph to save space.

---

> ### Author Response · Authors · 2021-11-18
> **Checking in**
>
> Hi. We again thank the reviewers for their detailed comments. Since the discussion period will end soon after the weekend, we would just like to check in and see if we can further clarify any of our responses or address any further concerns.

---

### Author Response · Authors · 2021-12-03
**Please give our comments and revision consideration**

Dear Reviewers,

It has been more than two weeks since we uploaded a much improved version of the paper and responded to much of your concerns. It seemed the most important concerns were real-world examples of how SSX could be used beyond the games we've experimented with, questions on scalability, and clarifications of terminology in the paper. We've also added quantitative experiments illustrating consistency and faithfulness properties and considered another baseline in the Appendix. Please give our comments and revision consideration as we have put considerable effort in creating them.

Thank you,

Authors

---

### Decision · Program_Chairs · 2022-01-20

**Decision:**

Reject

**Comment:**

This paper presents a new perspective for understanding reinforcement learning policies based on meta-states, as an effort to improve the explainability of RL control policies. After reviewing the revised paper and reading the comments from the reviewers, here are my comments:

- The paper is well-written and very concise.
- The strategy is novel and deserves merit.
- The utility of the explanation is not well described.
- The main concerns of the proposal are the utility of the explanation (that is not well described) and its usage in large discrete state spaces or continuous state spaces domains.

From the above, it is difficult to see the contribution and applicability of the paper in a clear manner.